# Nitrous oxide activates layer 5 prefrontal neurons via SK2 channel inhibition for antidepressant effect

Joseph Cichon [1,2] ✉, Thomas T. Joseph [1], Xinguo Lu[3,4], Andrzej Z. Wasilczuk [1], Max B. Kelz [1,2], Steven J. Mennerick[3,4], Charles F. Zorumski[3,4] & Peter Nagele [5]

Nitrous oxide ($N_2O$) induces rapid and durable antidepressant effects. The cellular and circuit mechanisms mediating this process are not known. Here we find that a single dose of inhaled $N_2O$ induces rapid and specific activation of layer V (L5) pyramidal neurons in the cingulate cortex of rodents exposed to chronic stress conditions. $N_2O$-induced L5 activation rescues a stress-associated hypoactivity state, persists following exposure, and is necessary for its antidepressant-like activity. Although NMDA-receptor antagonism is believed to be a primary mechanism of action for $N_2O$, L5 neurons activate even when NMDA-receptor function is attenuated through both pharmacological and genetic approaches. By examining different molecular and circuit targets, we identify $N_2O$-induced inhibition of calcium-sensitive potassium (SK2) channels as a key molecular interaction responsible for driving specific L5 activity along with ensuing antidepressant-like effects. These results suggest that $N_2O$-induced L5 activation is crucial for its fast antidepressant action and this effect involves novel and specific molecular actions in distinct cortical cell types.

Major depression is a heterogenous condition that diminishes psychosocial functioning and quality of life. Despite advances in understanding pathophysiology and antidepressant mechanisms, up to one-third of patients have failed responses to conventional treatments[1,2], possibly reflecting different subtypes affecting distinct cell types and neuronal networks[3]. Prefrontal cortical (PFC) circuit dysfunction remains most consistently involved with depression severity tied to deficits in neuronal activity, plasticity, and structure[4–7]. Both rodent models of chronic stress and depressed human patients show hypoactive PFC networks[4,8,9], impaired plasticity[10], and reduced synaptic number and brain volume[11–13]. Existing treatments for depression reverse many disease-associated circuit defects but often have slow-onset[3,14]. Psychedelics and anesthetics, such as ketamine and

nitrous oxide ($N_2O$)[15–17], produce both rapid and sustained antidepressant effects following a single treatment in patients suffering from treatment-resistant depression (TRD). How these novel antidepressants enact their rapid and durable antidepressant effects is poorly understood.

The success of ketamine as a rapid antidepressant reawakened interest in $N_2O$ as a possible antidepressant[18,19]. Two randomized controlled, early-phase clinical trials of $N_2O$ (25-50%) demonstrated promising results where TRD patients experienced rapid and sustained symptomatic relief, mirroring the rapid antidepressant effects of ketamine[17,18]. Like ketamine, $N_2O$'s molecular mechanism on neuronal function is thought to involve NMDA-receptor (NMDA-R) antagonism[20,21], but this has never been directly tested in vivo. It

[1]Department of Anesthesiology and Critical Care, Perelman School of Medicine, University of Pennsylvania, Philadelphia, PA, USA. [2]Mahoney Institute for Neurosciences, Perelman School of Medicine, University of Pennsylvania, Philadelphia, PA, USA. [3]Department of Psychiatry, Washington University School of Medicine, St. Louis, MO, USA. [4]The Taylor Family Institute for Innovative Psychiatric Research, Washington University School of Medicine, St. Louis, MO, USA. [5]Department of Anesthesia and Critical Care, University of Chicago, Chicago, IL, USA. ✉e-mail: joseph.cichon@pennmedicine.upenn.edu

remains unclear how $N_2O$ could drive rapid and durable therapeutic effects despite fast elimination from the brain (~5 min) and no metabolites[22]. Here, we investigated the cellular and circuit basis for $N_2O$'s antidepressant effect by imaging PFC microcircuits before and after $N_2O$ treatment.

## Results

### $N_2O$'s antidepressant-like response arises from rapid activation of L5 pyramidal neurons

Chronic stress is one important risk factor for depression[23]. In animal models, chronic stress recapitulates key features of the depressed brain, including maladaptive changes in neuronal structure, function, and behaviors[24]. In this work, we stressed mice with two different strategies, (1) chronic corticosterone (referred to as CORT) in drinking water[25,26] and (2) chronic exposure of a male C57BL/6 mouse to a male CD-1 aggressor mouse (termed chronic aggressor interactions or CAI), a variation on repeated social defeat stress (Fig. 1a, Supplementary Fig. 1a–c)[27,28]. Following these protocols, animals displayed an anxio-depressive-like state as evidenced by prolonged immobility time in tail suspension test (TST), decreased preference for sucrose in sucrose preference test, and reduced activity in open arms in the elevated plus maze as compared to controls exposed to daily handling (Fig. 1b–d). To determine if $N_2O$ exhibits rapid antidepressant-like effects in rodents exposed to stress, we administered inhaled $N_2O$ (50%) or $O_2$ (control; 100%) for 1 h to head-restrained mice via a nose cone (Fig. 1a, Supplementary Fig. 1d). We chose a 50% $N_2O$ dose instead of 25% because of the open-circuit delivery method in rodent studies, unlike the semi-closed-circuit approach commonly used in human trials. We found that $N_2O$ drove the rapid (within 1 h from treatment) reversal of stress associated behaviors in both CORT and CAI mice compared to $O_2$-treated controls (Fig. 1b–d). In a separate cohort of mice tracked only in TST we observed a sustained decrease in immobility time at 24 h post treatment in both CORT and CAI mice but not in controls (Supplementary Fig. 1f). Although it is difficult to assess drug-induced behavioral effects in head-restrained mice, unrestrained mice in a closed chamber exposed to 50% $N_2O$ showed increased movement and exploration rather than signs of sedation (Supplementary Fig. 2).

Dysfunction of the medial PFC is a hallmark feature of human depression[6] and preclinical models of chronic stress[29,30]. To determine if chronic stress produces a similar effect in rodent brain, we recorded the spontaneous activity of excitatory neurons across the cortical column in the cingulate cortex (Cg1), which is a supplemental motor area in rodents, by using two-photon (2-P) calcium imaging (Fig. 1a, right). Adeno-associated virus (AAV) encoding the calcium indicator GCaMP6f under the Calcium/Calmodulin dependent protein kinase II (CaMK2) promoter drove stable sensor expression specific to pyramidal neurons of superficial, i.e. layer (L) 2/3, and deep, L5, layers in this location (Supplementary Fig. 3c). Deep neurons of stressed mice (CORT and CAI) showed a hypoactivity state in L5 as compared to controls (Fig. 1e, g), a finding consistent with this region reconfiguring to chronic stress[8]. Thus, in agreement with previous reports[29], chronic stress in adult rodents induces both behavioral and neurophysiological changes consistent with a depression-like state.

Considering that $N_2O$ is believed to reduce both pre- and postsynaptic neuronal activity as a sedative-hypnotic with neuroprotective properties[20,31], it was unclear how $N_2O$ would produce its rapid antidepressant effect in a stress-induced hypoactive cortical network. Here, inhaled $N_2O$ (50%) induced rapid (within minutes) activation of deep pyramidal neurons (Avg. cell location ± std. error of mean, control: $573 ± 17\,\mu m$, CORT: $623 ± 14\,\mu m$, CAI: $619 ± 16\,\mu m$) in both chronically stressed and control mice as compared to both superficially located L2/3 neurons (control: $315 ± 20\,\mu m$, CORT: $297 ± 22\,\mu m$, CAI: $272 ± 24\,\mu m$) and $O_2$ controls from the same region (Fig. 1e–g, Supplementary Fig. 4a, d, g, Supplementary Movie 1). Comparisons were made after 15 min of $N_2O$ exposure to ensure steady state gas

concentration. Considering Cg1's anatomical location and cytoarchitecture, we proposed the deep pyramidal neurons recruited by $N_2O$ are likely within L5. L5 identity was further supported by calcium imaging in specific L5 transgenic Cre lines, where $N_2O$ recruited Rbp4 and Tlx3 expressing cells, but not Colgalt2 (Fig. 1h)[32]. Furthermore, calcium imaging of CaMK2-expressing cells at a depth of $>500–600\,\mu m$ from the pial surface in secondary motor cortex (Supplementary Fig. 4b) and primary somatosensory cortex (Supplementary Fig. 4b, c) also revealed putative L5 neuronal activation as compared to L2/3. Therefore, we refer to this $N_2O$-recruited deep neuronal population as L5 hereon.

Within a local imaging field of view, $N_2O$-active L5 cells generated large asynchronous calcium transients, resembling a form of burst-like firing (Supplementary Fig. 4e–g)[33]. Given the sustained nature of these transients, we show calcium activity as area under the curve (AUC) of the $\Delta F/F_0$ trace for an individual neuron, reflecting changes in both frequency and amplitude (Supplementary Fig. 4h). A dose-response curve of inhaled $N_2O$ concentrations revealed a strong activation CaMK2-expressing L5 population in Cg1 across all dosing with a peak effect at 50% $N_2O$ (Fig. 1i, Supplementary Fig. 1e). Electroencephalogram (EEG) recordings over the same $N_2O$ concentration steps revealed higher frequency oscillations at 50% as compared to 25 or 75% (Supplementary Fig. 5). These results suggest that at doses found to have antidepressant effects in human patients suffering from TRD[17,18], $N_2O$ inhalation drives a rapid activation of the same class of neurons most profoundly affected by chronic stress in rodents (Fig. 1g)[29].

Because $N_2O$ activates L5 from its baseline hypoactive state in chronically stressed mice, we wondered if this activation plays a pivotal role in $N_2O$'s antidepressant response. To specifically increase L5 activity in vivo (in absence of $N_2O$), we specifically transfected Rbp4 Cre-expressing mice with an AAV encoding Cre-dependent $hM_3D(G_q)$ designer receptor exclusively activated by designer drug (DREADD) receptor in Cg1 bilaterally (Fig. 1j, Supplementary Fig. 6a). The binding of the ligand clozapine N-oxide (CNO) to $hM_3D(G_q)$ receptors activates $G_q$-coupled signaling, leading to membrane depolarization and increased firing of target cells through multiple mechanisms[34]. CNO delivered by i.p. injection to mice expressing $hM_3D(G_q)$ specifically in L5 cells induced a ~2–3-fold increase in spontaneous calcium activity (when assessed at 1 and 3 hrs later) and a rapid antidepressant-like effect detected within 1 h that lasted at least for 1 day (Fig. 1j, Supplementary Fig. 6b-f). This behavioral effect induced by $hM_3D(G_q)$ in L5 masked $N_2O$'s effect when CNO was delivered prior to $N_2O$ (Fig. 1k). Conversely, L5 inactivation by DREADD-$hM_4D(G_i)$ acutely blocked $N_2O$'s ability to recruit L5 and its accompanying antidepressant effect when CNO was delivered prior to $N_2O$ (Fig. 1j, k, Supplementary Fig. 6d). These results suggest that a single $N_2O$ treatment induces rapid and specific activation of L5 neurons in Cg1 to rescue chronic stress-associated hypoactivity and that recruitment of these cells is required for $N_2O$'s ensuing antidepressant-like effect.

### $N_2O$-induced rescue of stress associated L5 hypoactivity persists following drug elimination

$N_2O$ is rapidly cleared from the brain/body within minutes (via expiration) without active metabolites[35]. To determine if $N_2O$'s acute L5 activation contributes to its lasting antidepressant-like effect, we followed the same L5 populations following $N_2O$ exhalation. To our surprise, we found persistent L5 activity in both chronically stressed and control mice 1 h following $N_2O$ exposure but not in $O_2$ treated mice (Fig. 2a, c, Supplementary Fig. 7a). L5 responses persisted for at least 3 h in chronically stressed mice as compared to $O_2$ treated control mice (Supplementary Fig. 7b). In some cases where L5 was tracked over 24 h, we found evidence of persistent activity (Supplementary Fig. 7d). At these follow-up time points, superficial L2/3 cells, previously weakly active or not active, were now found to be spontaneously active

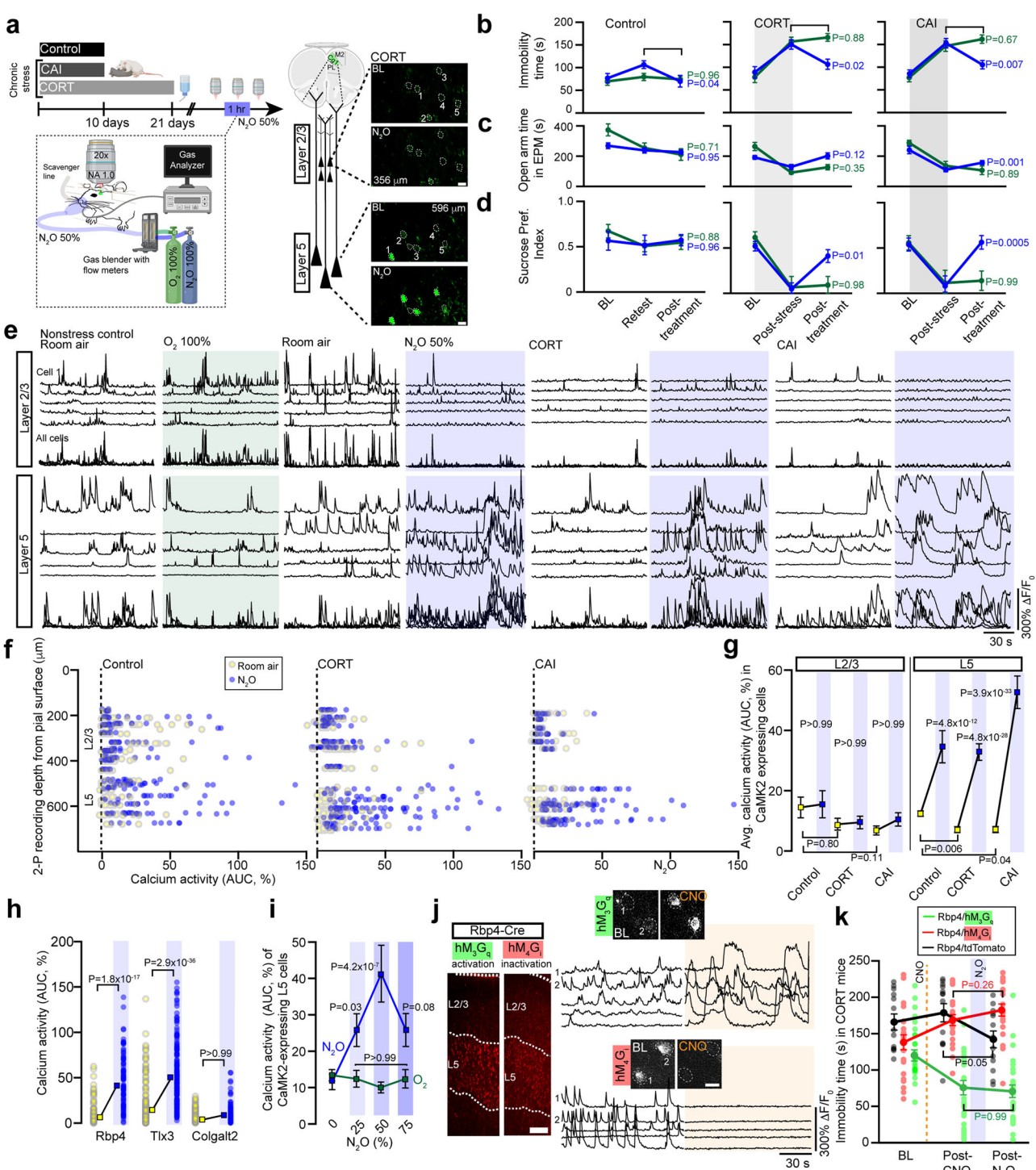

(Fig. 2b, c). Newly active L2/3 activity also persisted for hours (Fig. 2b, c, Supplementary Fig. 7d).

Given that L5 was recruited during $N_2O$ exposure and both L5 and L2/3 have persistent spontaneous activity post-treatment, we suspected that $N_2O$-L5 activation contributes to the recruitment and sustained activation of local L2/3 neurons in Cg1 (Fig. 2d). To determine if local L5 neuronal activity can drive local L2/3 activation, L5-specific Rbp4-Cre mice were transfected with AAVs encoding Cre-dependent $hM_3D(G_q)$ and synapsin-GCaMP6f for DREADD-induced L5 activation while recording L2/3 calcium responses respectively (Fig. 2e, left schematic). CNO-induced L5 activation drove L2/3 activation (2.5-fold increase in spontaneous activity) 20-30 min following CNO injection (Fig. 2f, g). To ascertain if post-$N_2O$ exposure L5 activity drives local L2/3 activity in chronically stressed mice, CORT-treated Rbp4 mice expressing $hM_4D(G_i)$ (unilateral expression) were exposed to $N_2O$ to drive L5 activation and persistent activity. L5 neurons were then chemogenetically inactivated by $hM_4G_i$, meanwhile recording ipsilateral L2/3 neuronal activity (Fig. 2e, right schematic). Following CNO injection (imaging 20–30 min after CNO), post $N_2O$-induced L2/3 activity was significantly reduced by L5 inactivation (Fig. 2f, g). We suspected the residual L2/3 activity (~34% increase over baseline) following local L5 silencing could be explained by long-range L5 projections (both ipsilateral and contralateral) into Cg1 (Fig. 2e). In agreement with this hypothesis, bilateral inhibition of Cg1 L5 neurons

**Fig. 1 | $N_2O$ induces rapid and specific activation of L5 pyramidal neurons to drive antidepressant-like response. a** Left, chronic stress was induced with either corticosterone in tap water (CORT) or exposure to screened aggressive male CD-1 mouse (chronic aggressor interactions or CAI). CORT, CAI, and control (exposed to daily handling) mice were subjected to two-photon calcium imaging in the Cg1 (imaging location denoted by green highlighted region with dashed line to cell types) before, during, and after nitrous oxide ($N_2O$) exposure. $N_2O$ was blended, delivered, and monitored under the microscope at 50% for 1 h. Right, CaMK2-expressing excitatory neurons located in layer 2/3 (L2/3) and layer 5 (L5) in CORT mice. Top (room air) and bottom ($N_2O$) images show 5 cells from each layer with their fluorescent transients from a 2 min time-series movie collapsed into a single image. Scale bar, 20 μm. Created in BioRender. Cichon, J. (2025) https://BioRender.com/d16f930. **b**–**d** Chronic stress (CORT, $N_2O$ $n = 16$, $O_2$ $n = 12$; CAI, $N_2O$ $n = 11$, $O_2$ $n = 10$) increased the time spent immobile in tail suspension test (**b**) avg. immobility time was $153 \pm 7$ seconds in CORT and $149 \pm 8$ s in CAI vs. $86 \pm 7$ s in controls; Kruskal-Wallis (33): $P = 5.6 \times 10^{-8}$ followed by Dunn's multiple comparisons, control vs. CORT: $P = 4.1 \times 10^{-7}$, control vs. CAI: $P = 6.8 \times 10^{-6}$, decreased exploration of open arms in an elevated plus maze (EPM) (**c**) avg. open arm time was $113 \pm 11$ s in CORT and $115 \pm 16$ s in CAI vs. $238 \pm 19$ seconds in control mice; Kruskal-Wallis (31): $P = 1.6 \times 10^{-7}$ followed by Dunn's multiple comparisons, control vs. CORT: $P = 1.3 \times 10^{-6}$, control vs. CAI: $P = 1.0 \times 10^{-5}$, and reduced sucrose preference index (SPI) (**d**), avg. SPI was $0.06 \pm 0.1$ in CORT and $0.1 \pm 0.1$ in CAI vs. $0.5 \pm 0.1$ in control mice; Kruskal-Wallis (22): $P = 1.6 \times 10^{-7}$ followed by Dunn's multiple comparisons, control vs. CORT: $P = 2.5 \times 10^{-5}$, control vs. CAI: $P = 8.7 \times 10^{-4}$, as compared to control mice ($N_2O$ $n = 14$, $O_2$ $n = 14$). $N_2O$, but not $O_2$, therapy rapidly reversed the effects of chronic stress (2-way ANOVA time x treatment: TST CORT, $F_{(2, 78)} = 8.1$, $P = 8.6 \times 10^{-4}$, TST CAI, $F_{(2, 38)} = 7.4$, $P = 0.002$; EPM CORT, $F_{(2, 78)} = 6.5$, $P = 0.002$, EPM CAI, $F_{(2, 38)} = 3.0$, $P = 0.06$; SPI CORT, $F_{(2, 52)} = 4.0$, $P = 0.02$, SPI CAI, $F_{(2, 57)} = 7.6$, $P = 0.001$). Post treatment comparisons ($N_2O$, blue; $O_2$, green) are shown in panel (**e**), Representative GCaMP6 traces of the spontaneous activity of individual neurons from L2/3 and L5, shown in (**a**), under wakefulness followed by oxygen (left, green) or $N_2O$ (right, blue) in control and chronically stressed mice.

$N_2O$ induced the rapid recruitment of L5 neurons as compared to L2/3 across all conditions. **f** Individual neuronal responses (circles) under room air (yellow) and $N_2O$ (blue) from all recording regions across Cg1 from control (left) and chronically stressed (middle/CORT and right/CAI) mice. Oxygen plot in Fig. S4a. **g** L2/3 and L5 population response (colored squares) under room air and $N_2O$ across control (L2/3: $n = 121$; L5: $n = 102$ from 8 mice) and CORT (L2/3: $n = 96$; L5: $n = 122$ from 9 mice)/CAI mice (L2/3: $n = 61$; L5: $n = 98$ from 4 mice). $N_2O$ drove the rapid L5 activation across stressed and control mice (Kruskal-Wallis (390): $P = 5.3 \times 10^{-77}$ followed by Dunn's multiple comparisons, control: $P = 4.8 \times 10^{-12}$, CORT: $P = 4.8 \times 10^{-28}$, CAI: $P = 3.9 \times 10^{-33}$) as opposed to L2/3 (Kruskal-Wallis: control/CORT/CAI: $P > 0.99$). L5 neurons from chronically stressed mice displayed a hypoactivity state relative to control mice (from prior Dunn's test: CORT, $P = 0.006$; CAI, $P = 0.04$). L2/3 activity was also reduced but not significant (CORT, $P = 0.80$; CAI, $P = 0.11$). **h** L5 calcium responses under room air and $N_2O$ across different genetically defined L5 neuronal subtypes (Kruskal-Wallis (404): $P = 4.4 \times 10^{-85}$ followed by Dunn's multiple comparisons, Rbp4 ($n = 89$ from 3 mice): $P = 1.8 \times 10^{-17}$, Tlx3 ($n = 196$ from 4 mice): $P = 2.9 \times 10^{-36}$; Colgalt2 ($n = 149$ from 4 mice): $P > 0.99$). **i** CaMK2-expressing L5 neuronal responses at different $N_2O$ concentrations (0, 25, 50, 75% mixed with $O_2$). L5 activating effect observed at 25% with peak effect at 50% (Two-way ANOVA with Bonferroni's comparisons: 25%, $P = 0.03$; 50%, $P = 4.2 \times 10^{-7}$; 75%, $P = 0.08$). **j** Left, coronal sections of Cg1 from Rbp4-Cre mouse expressing either Cre-dependent chemogenetic variant (tagged with mCherry) $hM_3G_q$ (green) or $hM_4G_i$ (red; Scale bar 100 μm). Right, representative GCaMP6 traces (bottom) and two-photon images (top) of individual L5 neurons recorded during room air and following CNO injection (orange shared area). Scale bar, 20 μm. **k** CNO-induced Rbp4-L5 neuronal inactivation with $hM_4G_i$ bilaterally in Cg1 ($n = 22$) blocked $N_2O$'s effect on TST whereas CNO-induced Rbp4-L5 activation with $hM_3G_q$ ($n = 14$) masked $N_2O$'s effect (two-way ANOVA time x treatment $F_{(4, 166)} = 19.8$, $P = 1.9 \times 10^{-12}$) post hoc Sidak's comparisons shown in panel following $N_2O$ exposure. $N_2O$, but not CNO, reduced immobility times in Rbp4 mice expressing tdtTomato ($n = 8$). Representative images and traces carried out on at least three animals per group. Error bars show s.e.m.

---

post-$N_2O$ further reduced spontaneous L2/3 activity (-18% decrease from baseline) (Fig. 2f, g). Furthermore, bilateral inhibition of both L2/3 and L5 activity post-$N_2O$ via CNO-induced $hM_4G_i$ signaling promoted the reversal $N_2O$-induced antidepressant effect (Supplementary Fig. 7c). These experiments demonstrate that $N_2O$ exposure drives the rapid and persistent activity of L5 neurons. Following $N_2O$ elimination, L5 neurons contribute to the recruitment of excitatory neurons in interconnected circuits, which contributes $N_2O$'s antidepressant-like response.

## Reduced NMDA-R function does not block $N_2O$-induced L5 activation

$N_2O$-induced L5 rapid and persistent activity is seemingly at odds with $N_2O$'s known molecular mechanism as an NMDA-R antagonist as NMDA-R block would likely attenuate neuronal activity[36–38]. To determine if NMDA-R activity is required for $N_2O$-induced L5 rapid and persistent activity, we recorded L5 responses before and after local application NMDA-R antagonists (both D-APV and MK801) delivered to Cg1 through a small bone hole lateral to the imaging region[39,40], and once again in the presence of $N_2O$ (Fig. 3a). Consistent with the expected effects of potent NMDA-R antagonists, both D-APV and MK801 (100 μM, 1 μL) suppressed spontaneous L5 calcium activity. $N_2O$, however, retained its ability to recruit L5 activity even in the presence of NMDA-R blockers (Fig. 3b, d, Supplementary Fig. 8a). Lower concentrations of MK801 (10 μM, 1 μL) yielded a similar $N_2O$-induced effect despite increasing baseline L5 activity (Fig. 3b, d). Furthermore, L5 activity even persisted following local co-application of potent excitatory synaptic blockers - MK801 and AMPA receptor blocker CNQX (each 100 μM, total volume 1 μL; Fig. 3b, d). Thus, in the presence of either NMDA-R blockers or excitatory synaptic blockers, $N_2O$ retains its ability to rapidly activate L5.

When L5 neurons were followed post-$N_2O$ exposure, NMDA-R blockers significantly reduced $N_2O$'s effect on persistent L5 activity (Supplementary Fig. 8a). Similarly, in mice first treated with $N_2O$ and then exposed to local NMDA-R blocker, $N_2O$-induced L5 persistent activity was significantly dampened (Supplementary Fig. 8a, b). Therefore, L5 activation and persistent activity could be manifested by two distinct mechanisms: (1) an unknown, NMDA-R-independent activation mechanism, (2) persistent activity maintained by NMDA-R-dependent activity. In support of this claim, $N_2O$ inhalation drove L5 activation in the presence of NMDA-R subunit 1 (NR1) expression knockdown via siRNA targeted to GRIN1, but with a significant impairment in persistent activity at 1 h (Fig. 3c, d, Supplementary Fig. 8c). Scrambled siRNA showed no impairments in L5 activity during or post $N_2O$ exposure (Fig. 3c, d). Thus, contrary to its proposed mechanism of action through NMDA-R antagonism, our findings show that $N_2O$ rapidly activates L5 neurons even when NMDA-R function is diminished.

Similar to $N_2O$, ketamine is believed to exert its rapid and long-lasting antidepressant effects primarily through NMDA receptor antagonism[15,16]. Given that $N_2O$-induced activation of L5 neurons occurs even with reduced NMDA receptor function, we hypothesized that ketamine and $N_2O$ would modulate the spontaneous activity of local Cg1 L5 neurons in distinct ways. To this end, we followed L5 populations through different iterations of $N_2O$ and subhypnotic ketamine exposures and found L5 neurons displayed opposing modulation by the two drugs (Fig. 3e–g). While $N_2O$ drove the stronger L5 response between the two treatments, ketamine-activated L5 neurons showed little overlap with $N_2O$-activated cohort (Fig. 3g). Local application of ketamine (100 μM, 1 μL) failed to block or recapitulate $N_2O$-induced L5 activation (Fig. 3e, f). When systemic ketamine was delivered before or in between two $N_2O$ exposures, ketamine inhibited the succeeding $N_2O$-induced L5 activation (Fig. 3e, f). Therefore, $N_2O$ and ketamine have highly divergent modulations of L5 activity. This implies both drugs likely engage different cellular and circuit mechanisms to achieve their acute antidepressant effects.

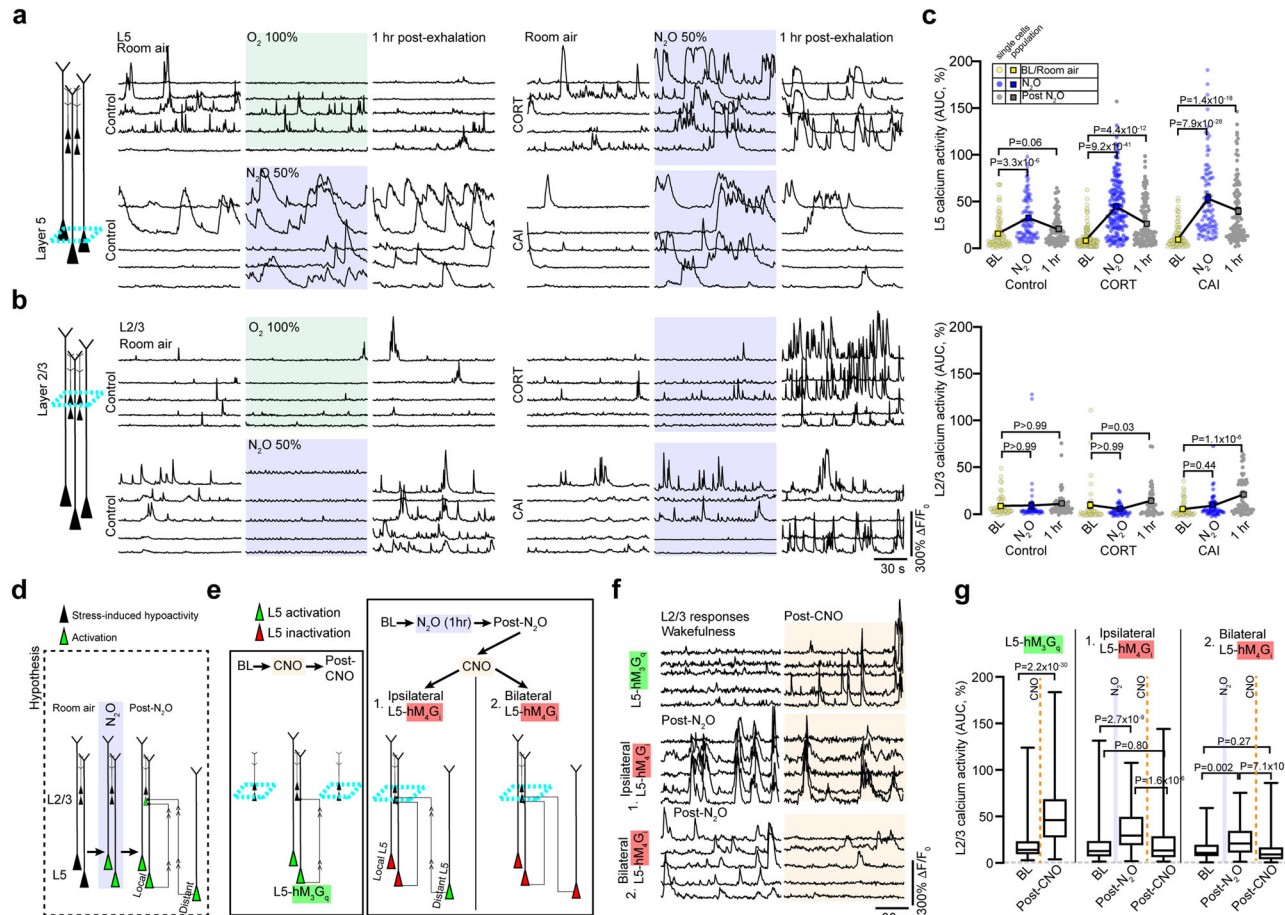

**Fig. 2 | Persistent L5 activity following N$_2$O elimination contributes to L2/3 recruitment. a**, **b** The same individual L5 (**a**) and L2/3 (**b**) neuronal responses under room air (left), O$_2$, or N$_2$O (middle), 1 h after O$_2$ or N$_2$O (right) in control, CORT-treated and CAI mice. Note the persistent activity of L5 neurons and recruitment of L2/3 neurons following N$_2$O. **c** Summary of individual (circles) and population (colored squares connected by black line) neuronal responses of both L5 and L2/3 neurons across room air (yellow), N$_2$O exposure (blue), and following N$_2$O exhalation (gray) in control (L2/3: n = 52; L5: n = 88 from 3 mice) and CORT (L2/3: n = 43; L5: n = 163 from 5 mice)/CAI (L2/3: n = 61; L5: n = 97 from 4 mice) mice. Cohort different from Fig. 1. Persistent activation of L5 was observed at 1 h following N$_2$O exposure in all groups (Kruskal-Wallis (559): P = 5.5 × 10$^{-108}$ followed by Dunn's multiple comparisons, control, P = 0.06; CORT, P = 4.4 × 10$^{-12}$; CAI, P = 1.4 × 10$^{-18}$) while significant L2/3 activation was observed in only CORT-treated and CAI groups at 1 h following N$_2$O exposure (prior Dunn's test: control, P > 0.99; CORT, P = 0.03; CAI, P = 1.1 x 10$^{-6}$). **d** Schematic illustrating that hypoactive L5 neurons (black cells) become rapidly activated by N$_2$O treatment (green cells). Following N$_2$O exhalation, persistent L5 activity, from local or distant regions (green activated cells), may

recruit L2/3 (superficial green cells receiving projections). **e** Experimental design to test whether (1) direct local chemogenetic L5 activation results in recruitment of local L2/3 neurons (left), (2) Post-N$_2$O recruitment of L2/3 is dependent upon persistent local and distant L5 activity. **f** Individual L2/3 neurons responses corresponding to (**e**). Top, L2/3 responses before and after CNO-induced activation of L5 neurons expressing hM$_3$G$_q$ (n = 161 from 5 mice). Middle and bottom traces, L2/3 responses following either ipsilateral (n = 83 from 3 mice) or bilateral (n = 77 from 3 mice) hM$_4$G$_i$-mediated L5 inhibition respectively. **g** Summary of individual L2/3 responses from (**e**). Box plots (extending from 25th to 75th percentile with median in the middle and whiskers denoting min to max values) show CNO-induced activation of L5-hM$_3$G$_q$ drove superficial L2/3 activity (two-sided Wilcoxon matched-pairs signed rank test: P = 2.2 × 10$^{-30}$). Either ipsilateral (Kruskal-Wallis (42.6): P = 5.7 x 10$^{-10}$ followed by Dunn's multiple comparisons, post-N$_2$O: P = 2.7 x 10$^{-9}$; post-CNO: P = 0.80) or bilateral (Kruskal-Wallis (26.9): P = 1.4 × 10$^{-6}$ followed by Dunn's multiple comparisons, post-N$_2$O: P = 0.002; post-CNO: P = 0.27) CNO-induced L5-hM$_4$G$_i$ inhibition reduced post-N$_2$O-induced L2/3 activity to baseline wakefulness. Cells/animals per condition listed in (**f**). Error bars show s.e.m.

N$_2$O-induced L5 recruitment in the presence of synaptic blockers suggests a synaptic-independent mechanism. To specifically address this hypothesis, we performed dendritic imaging across individually labeled L5 neurons (different cohort from mice in Figs. 1–3). Using a sparse AAV labeling approach, individual L5 neurons expressing both GCaMP6 and tdTomato (structural marker) can be mapped within an imaging window from apical dendritic tree down to L5 soma (Fig. 4b). In dendritic segments confined to layer 1 (63 ± 8.3 μm), N$_2$O did not significantly increase the spontaneous activity of postsynaptic dendritic spines nor the generation of dendritic branch calcium events (Fig. 4a–c, Supplementary Fig. 9a). Deeper imaging of dendritic branch points (tuft: 100 ± 7.6 μm) also revealed no significant elevation in spontaneous calcium events comparable to baseline (Fig. 4b, c). L2/3 neurons were occasionally labeled with this technique (316 ± 21 μm) were not

recruited by N$_2$O (Supplementary Fig. 9b). Imaging at the location of L5 soma (627 ± 15 μm) and trunks (333 ± 31 μm) demonstrated robust N$_2$O-induced activation (Fig. 4b, c, Supplementary Fig. 9a).

While high-resolution imaging of dendritic segments captures a small portion of the L5 neuron's apical tuft at a given time (Fig. 4a, b), it is conceivable that other dendritic branches are active and contributing top-down inputs that were missed. To address this possibility, we performed two-photon laser directed apical tuft dendritic cuts[41], or dendritomies, from L5 cells during N$_2$O administration and recorded the impact on L5 responses. N$_2$O still drove L5 activation despite apical dendrites being physically separated from their soma (Fig. 4d, e, Supplementary Fig. 9c). Furthermore, in an in vitro low-density primary cortical neuronal culture (free of network variables), where synaptic connectivity is reduced (no detectable spontaneous

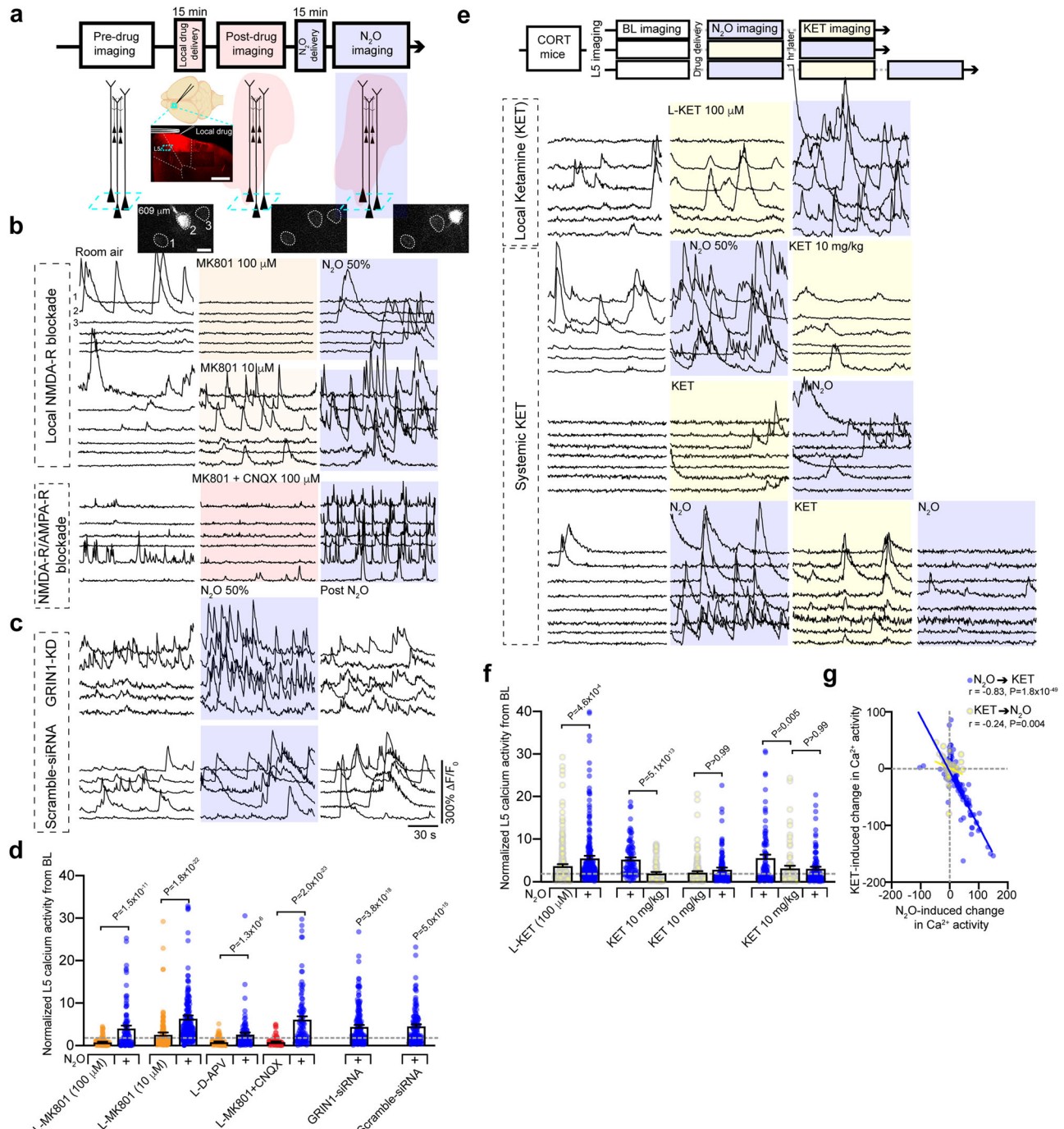

**Fig. 3 | Reduced NMDA-receptor activity does not impair N₂O-induced L5 activation in vivo. a** Top, timeline of L5 imaging (teal box) before, following local (L-) drug application (red cloud), and during N₂O exposure (blue shaded region). Middle, cartoon and representative coronal section of Cg1 region after pressure application of MK801 mixed with Rhodamine 6 G. Scale bar, 500 μm. Created in BioRender. Cichon, J. (2025) https://BioRender.com/i08h748. Bottom, Representative two-photon images of peak GCaMP6 signal from Cg1 L5 neurons under room air conditions, L-MK801, N₂O. **b** Representative GCaMP6 traces of L5 neurons under room air conditions, L-MK801 (top−high concentration/100 μM; $n = 103$ from 3 mice; middle−low concentration/10 μM; $n = 202$ from 4 mice), or L-MK801 + CNQX (lower traces; each drug at 100 μM mixed; $n = 96$ from 3 mice) followed by N₂O exposure. **c** Representative GCaMP6 traces of L5 neurons from mice infected with siRNA specific to GRIN1 (top; $n = 165$ from 4 mice) and scramble-siRNA (bottom; $n = 116$ from 4 mice) under room air, following N₂O. **d** Summary of individual L5 responses (from **b, c**) normalized to its baseline captured under room air conditions (Kruskal-Wallis (513): $P = 4.3 \times 10^{-103}$ followed by

Dunn's multiple comparisons, L-MK801 100 μM, $P = 1.5 \times 10^{-11}$; L-MK801 10 μM, $P = 1.8 \times 10^{-22}$; L-APV, $P = 1.3 \times 10^{-6}$; L-MK801 + CNQX, $P = 2.0 \times 10^{-23}$; GRIN1-siRNA, $P = 3.8 \times 10^{-18}$; Scramble siRNA, $P = 5.0 \times 10^{-15}$. Cells/animals per condition listed in (**b, c**). **e** L5 imaging of CORT mice exposed different drug sequences. Top, room air, L-ketamine (100 μM; $n = 243$ from 4 mice), followed by N₂O. Middle 2 traces, room air, N₂O exposure, followed by systemic ketamine (10 mg/kg i.p. injection; $n = 88$ neurons from 3 mice) and its reverse order (ketamine then N₂O; $n = 140$ neurons from 4 mice). Bottom, room air → N₂O → systemic ketamine → N₂O re-exposure ($n = 104$ neurons from 3 mice). **f** Summary of L5 responses from **e** (Kruskal-Wallis (135): $P = 2.5 \times 10^{-25}$ followed by Dunn's multiple comparisons, L-ketamine → N₂O, $P = 4.6 \times 10^{-4}$; N₂O → ketamine, $P = 5.1 \times 10^{-13}$; ketamine → N₂O, $P > 0.99$; N₂O → ketamine (comparison underlined) → N₂O, $P = 0.005$; N₂O → ketamine → N₂O, $P > 0.99$). Cells/animals per condition listed in (**e**). **g** N₂O-induced activity was negatively correlated with ketamine-induced activity in L5 neurons (two-sided Pearson correlation: N₂O → ketamine, $r = -0.83$, $P = 1.8 \times 10^{-49}$; ketamine → N₂O, $r = -0.24$, $P = 0.004$). Error bars show s.e.m.

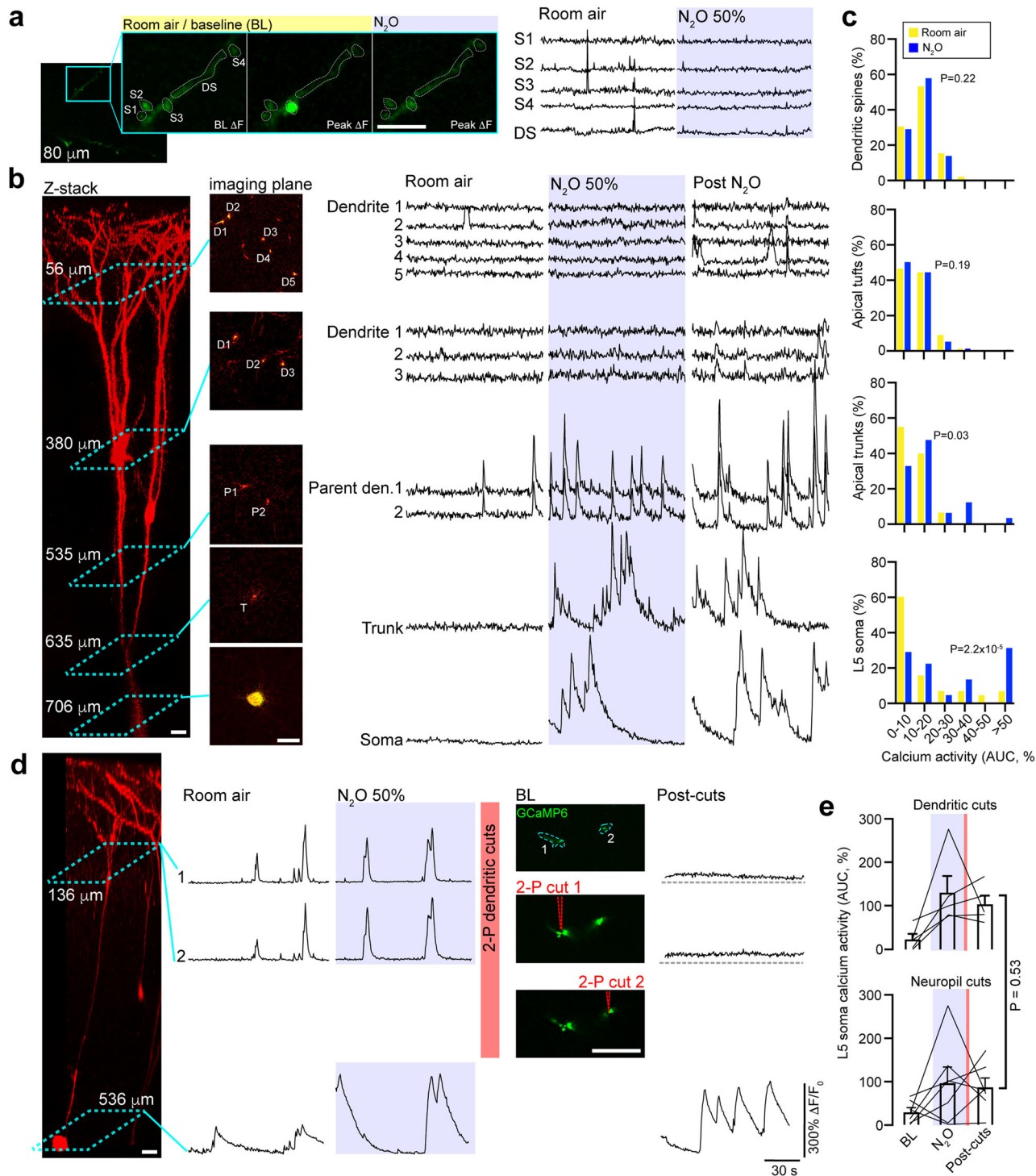

calcium activity prior to N$_2$O), N$_2$O-bubbled bath solution induced the rapid activation of a subset of putative pyramidal cortical neurons (Supplementary Fig. 10). Therefore, N$_2$O-induced L5 activation arises independently of NMDA-R activity and synaptic inputs and likely engages a novel, unknown somatic activation mechanism. Nevertheless, the persistent L5 activity observed following N$_2$O exposure requires NMDA-R activity.

## N$_2$O-induced cortical disinhibition promotes L5 activity and antidepressant-like effects

The absence of enhanced synaptic inputs or organized dendritic activity suggests N$_2$O is driving rapid changes in L5 excitability

through a novel somatic mechanism: either a molecular interaction, circuit reconfiguration, or a combination of the two. To this end, we evaluated several candidate channels and receptors that could hypothetically contribute to changes in L5 excitability by imaging L5 responses before and after local delivery of specific channel blockers followed by N$_2$O. Functional attenuation of voltage-gated sodium channels, voltage-gated calcium channels, serotonin reuptake transporters, mu opioid receptors, and channels controlling the intracellular release of calcium stores (both IP$_3$ and ryanodine) failed to block N$_2$O's rapid L5 activation (Supplementary Fig. 11a, b). Despite N$_2$O-induced L5 recruitment under these conditions, we did observe predicted changes in L5 calcium signal strength and

**Fig. 4 | N$_2$O fails to recruit synapses and dendrites to drive L5 activity.**
**a** Representative two-photon GCaMP6 image of L5 apical dendritic segment (DS; teal box) containing 4 spines (S1, S2, S3, S4) in 2-D imaging plane. Spontaneous spine activity (middle image showing peak signal of single spine activation) is occasionally observed in min recordings under room air conditions. N$_2$O fails to recruit spine activity ($n$ = 66 spines from 9 mice, two-sided Wilcoxon matched-pairs signed rank test: $P$ = 0.22). Scale bar, 5 μm. **b** Left, two-photon z-stack of an individual L5 neuron coexpressing GCaMP6 and tdTomato. Middle, multiple 2-D imaging planes across L5 neuron from apical tuft dendrites to soma corresponding to the teal boxes located on z-stack. ROI labeled D for apical dendrite, P for parent dendrite, T for trunk. Right, GCaMP6 traces for ROIs under room air, N$_2$O inhalation, and following N$_2$O exhalation. N$_2$O fails to recruit superficial dendrites during treatment. Dendritic activity is found deep, in close proximity to soma. Scale bar, 20 μm. **c** Summary of calcium responses from across different L5 dendritic compartments (dendritic segments: $n$ = 170 from 21 mice, two-sided Wilcoxon matched-pairs signed rank test: $P$ = 0.19; deep dendrites/trunks: $n$ = 34 from 14 mice,

Wilcoxon matched-pairs signed rank test: $P$ = 0.03) and soma ($n$ = 47 from 19 mice, Wilcoxon matched-pairs signed rank test: $P$ = 2.2 × 10$^{-5}$). **d** Left, Z-stack of L5 neuron coexpressing GCaMP6 and tdTomato subject to dendritomies by 2-photon laser pulses. Right, GCaMP6 traces of dendritic ROIs and soma under wakefulness, N$_2$O, and following dendritomies. L5 soma and dendrites were activated by N$_2$O as compared to wakefulness. GCaMP images of two parents dendrites corresponding to teal box in Z-stack (ROI 1 and 2 at 136 μm) are targeted and sequentially cut using laser pulses resulting in a baseline fluorescence bump coupled with the elimination of transients. L5 activity persists following dendritomies despite loss of dendritic activity. Scale bar, 20 μm. **e** Top, Summary of individual L5 calcium activity following N$_2$O exposure and apical dendritomies ($n$ = 5 L5 neurons from 5 mice). Bottom, L5 neurons under same conditions exposed to focal two-photon laser pulses directed at neuropil (control; $n$ = 7 L5 neurons from 7 mice). L5 responses following neuropil pulses were not significantly different than those directed at dendrites (two-sided Mann Whitney rank sum (13): $P$ = 0.53). Error bars show s.e.m.

duration imposed by the local drug indicating adequate diffusion to L5 (Supplementary Fig. 11a, b).

Next, we evaluated whether GABA receptor neuromodulation could regulate N$_2$O-induced activity. By taking advantage of the GABAergic volatile anesthetic gas isoflurane, which is often coadministrated with N$_2$O in clinical practice (e.g., operating room), we found that subhypnotic isoflurane concentrations, both 0.2% and 0.6%, mixed with N$_2$O (50%) blocked the rapid recruitment of L5 neurons (Supplementary Fig. 12a, c). Similarly, local application of a potent and selective GABA$_A$-R agonist muscimol induced a strong blockade of N$_2$O-evoked L5 activity (Supplementary Fig. 12b, c). The coadministration of isoflurane (0.6%) with N$_2$O (50%) also eliminated N$_2$O's antidepressant-like effect (Supplementary Fig. 12d). Thus, N$_2$O-induced L5 recruitment is highly sensitive to acute changes in GABA receptor-mediated inhibition and supports the conjecture that N$_2$O-induced L5 activity is necessary for its antidepressant-like effects.

While N$_2$O appears to interact weakly with postsynaptic GABAergic receptors[42], the action of N$_2$O on GABA-releasing interneurons is unknown. GABAergic interneurons target specific domains of pyramidal neurons and other local interneurons, providing precise control of excitatory and inhibitory outputs and cortical dynamics (Fig. 5a). A rapid shift in cortical inhibition could present one mechanism to explain N$_2$O's effect on pyramidal cell excitability. To determine if interneurons contribute to N$_2$O-induced L5 activity, first we mapped spontaneous activity of interneuron responses under room air and during N$_2$O exposure in mice expressing GCaMP6 under the m*Dlx* enhancer, a specific labeling strategy for GABAergic interneurons (Fig. 5b, c)[43]. We found that N$_2$O induced the overall suppression of interneurons activities from baseline wakefulness (110/137 cells) with only a small subset of cells becoming recruited by N$_2$O (Fig. 5c, Supplementary Fig. 13a). Three genetically defined subtypes: PV-, SST-, VIP-expressing interneurons are subsumed within the m*Dlx*-defined interneuron population (Fig. 5a, right cartoon). We suspected that the small population of m*Dlx* cells recruited by N$_2$O was specific to one of these subtypes. By taking advantage of several interneuron-specific Cre driver lines coupled with AAV transfection of Cre-dependent GCaMP6f, we examined the activity profiles of interneurons in Cg1 (Fig. 5b). Here, we found that N$_2$O-induced the downregulation of PV and SST activities from baseline measurements (Fig. 5c, Supplementary Fig. 12a). In contrast to PV and SST activities, N$_2$O increased VIP activity (Fig. 5c, Supplementary Fig. 12a). Therefore, N$_2$O induces the rapid reconfiguration of local interneurons favoring the establishment of a disinhibitory circuit, a motif akin to ketamine[39].

To assess whether the N$_2$O-induced downregulation of PV or SST activities is required for L5 activation, we attempted to counteract the suppression of interneuron activity induced by N$_2$O by autonomously activating these cells using DREADD variant hM$_3$D(G$_q$) (Supplementary Fig. 13b). To drive interneuron activity in vivo, CNO-injected mice

coexpressing GcaMP6f and hM$_3$D(G$_q$) specifically in PV, SST, and VIP-expressing cells induced more than two-fold increase in spontaneous calcium activity in wakefulness (Fig. 5d). CNO-mediated interneuron activation was maintained in the presence of N$_2$O (Fig. 5d). Next, we activated these interneuronal subtypes individually before N$_2$O while monitoring L5 neuronal activity. CNO-induced activation of either PV or SST interneurons prevented the N$_2$O-induced L5 activity (Fig. 5e). Similarly, if CNO was given after N$_2$O (instead of before), N$_2$O-induced L5 activation was quickly abolished (Supplementary Fig. 13d). By contrast, CNO-induced VIP activation did not prevent the N$_2$O-induced L5 activity (Fig. 5e). In control mice expressing tdTomato, CNO did not impair N$_2$O's L5 response (Supplementary Fig. 13c). Furthermore, the presence of N$_2$O-induced L5 activation (VIP activation) led to an antidepressant-like response where its absence (PV or SST activation) did not (Fig. 5f). Taken together these experiments suggest N$_2$O induces a disinhibitory circuit favoring an increase in L5 activity.

Such a dramatic shift in SST and PV-expressing populations should indiscriminately favor pyramidal cell activation (excitatory cells in L2/3 and L5) given known patterns of connectivity. While N$_2$O-induced disinhibition enables L5 activity it fails to account for N$_2$O's observed L5 specificity. Therefore, we reasoned that an unidentified specific molecular interaction in both VIP and L5 neurons could explain N$_2$O's cellular and circuit changes in cortex.

## SK2 channel inhibition reproduces L5 and VIP cell activation and antidepressant-like effects

N$_2$O's well described in vitro NMDA-R antagonism effects are unlikely to underlie the activation of both L5 and VIP cells in vivo (Fig. 3). We predicted N$_2$O's ability to recruit L5 and VIP cells would rest upon a shared mechanism in both cell types, given the lack of direct connectivity between the two cell types[44], which would regulate acute changes in excitability and depolarization to drive rapid and persistent L5 activity. To this end, we explored single-cell RNA expression levels of receptors and ion channels across genetically defined pyramidal and interneuron cell types using the open source Allen Brain Cell Atlas (Supplementary Fig. 14a)[45]. We uncovered a voltage-insensitive, calcium-sensitive potassium channel 2 (SK2 channel encoded by gene KCNN2) as a potential molecular target given: (1) its increased RNA expression in both L5 and VIP cells (Supplementary Fig. 14a), (2) SK2 protein immunohistochemistry and KCNN2 in situ hybridization identifying predominant L5 expression (Fig. 6a, Supplementary Fig. 14b)[46], (3) its role in regulating pyramidal cell intrinsic excitability and plasticity[47,48], (4) SK channel inhibition induces antidepressant effects in rodents[49,50].

First, we investigated the effects of N$_2$O on the medium after-hyperpolarization (mAHP), the major function of SK2 channels, in Cg1 L5 neurons in acute brain slices[51,52]. mAHPs were elicited by step-current injections ranging from 200 to 400 pA. Here, bubbled N$_2$O

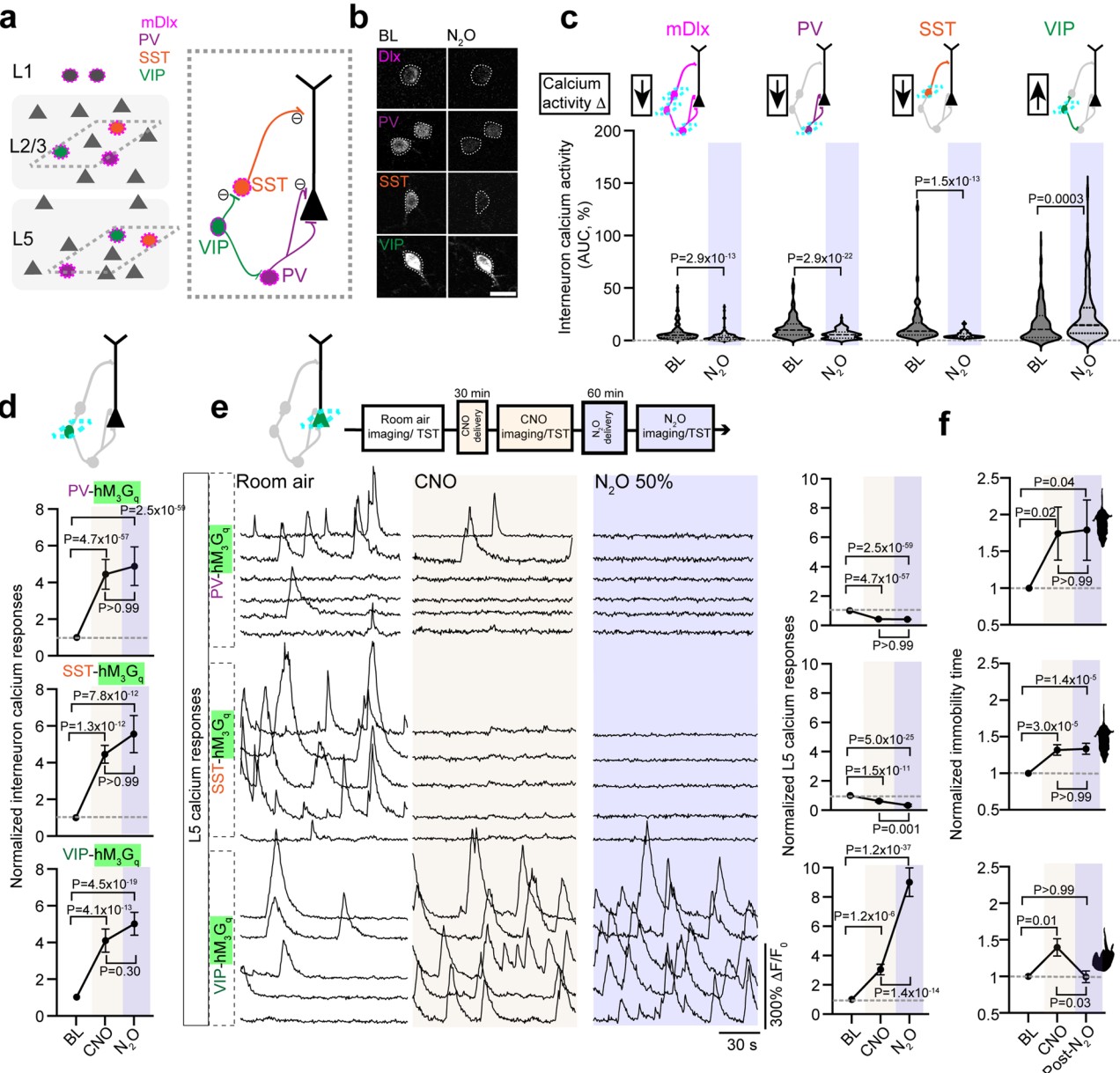

**Fig. 5 | N₂O-induced L5 activity and antidepressant-like effect requires cortical disinhibition. a** Left, Schematic of GABAergic neurons (circles) scattered amongst pyramidal neurons (triangles). While all interneurons can be labeled with GCaMP6 using the *mDlx* enhancer (outer magenta), subtypes including PV (purple), SST (orange), and VIP (green) can be specifically labeled using Cre transgenic lines depicted on right. Dendritic targeting SST cells and somatic targeting PV are inhibited (circled negative sign) by VIP interneurons. **b** Representative two-photon images of peak GCaMP6 signals from interneuron subtypes under room air and N₂O (50%). Scale bar, 20 μm. Traces in fig. S13. **c** Summary of genetically defined interneuron calcium responses under room air and N₂O. N₂O induces a suppression of PV and SST spontaneous activity but activates VIP activity (two-sided Wilcoxon matched-pairs signed rank: mDlx, $n = 133$ cells from 5 mice, $P = 2.9 \times 10^{-13}$; PV, $n = 145$ cells from 7 mice, $P = 2.9 \times 10^{-22}$; SST, $n = 63$ cells from 4 mice, $P = 1.5 \times 10^{-13}$; VIP, $n = 197$ cells from 10 mice, $P = 0.0003$. **d** Interneuron subtypes coexpressing GCaMP6 and DREADD-hM₃Gq recorded under wakefulness, post-CNO injection, and N₂O (50%). CNO-hM₃Gq induced activation of interneuron subtypes blocked N₂O induced suppression of PV ($n = 84$ cells from 3 mice; Kruskal-Wallis (347): $P = 3.0 \times 10^{-76}$ followed by Dunn's multiple comparisons, $P > 0.99$) and SST activity

($n = 103$ cells from 3 mice; Kruskal-Wallis (68): $P = 2.0 \times 10^{-15}$ followed by Dunn's multiple comparisons, $P > 0.99$). VIP cells displayed a similar trend ($n = 97$ cells from 4 mice; Kruskal-Wallis (93): $P = 6.8 \times 10^{-21}$ followed by Dunn's multiple comparisons, $P = 0.30$). **e** Left, representative GCaMP6 traces of individual L5 responses and summary of all cells (right) under room air, post-CNO injection, and N₂O. CNO-induced activation of PV ($n = 241$ cells from 7 mice; Kruskal-Wallis (348): $P = 3.0 \times 10^{-76}$ followed by Dunn's multiple comparisons, $P > 0.99$) or SST ($n = 110$ cells from 3 mice; Kruskal-Wallis (112): $P = 3.3 \times 10^{-25}$ followed by Dunn's multiple comparisons, $P = 0.001$) blocked N₂O-induced L5 activation whereas VIP promoted N₂O-induced L5 activity ($n = 139$ cells from 3 mice, Kruskal-Wallis (169): $P = 1.8 \times 10^{-37}$ followed by Dunn's multiple comparisons, $P = 1.4 \times 10^{-14}$). **f** TST immobility time under the same conditions. PV ($n = 13$, Kruskal-Wallis (9): $P = 0.01$ followed by Dunn's multiple comparisons, $P > 0.99$) and SST ($n = 17$, Kruskal-Wallis (27): $P = 1.4 \times 0^{-6}$ followed by Dunn's multiple comparisons, $P > 0.99$) activation by hM₃Gq blocked N₂O-induced decrease in immobility time in CORT mice. VIP activation prior to N₂O produced a significant decrease in immobility time ($n = 14$, Kruskal-Wallis (10): $P = 0.006$ followed by Dunn's multiple comparisons, $P = 0.03$). Error bars show s.e.m.

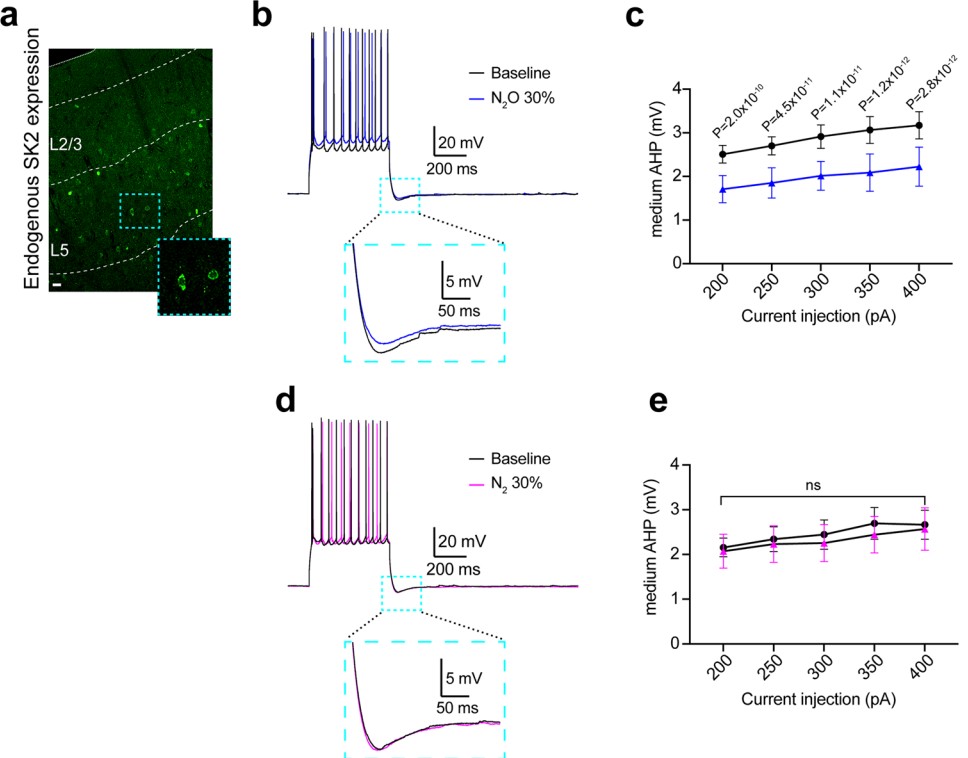

**Fig. 6 | $N_2O$ inhibits medium afterhyperpolarization potentials in L5 neurons.**
**a** Mouse Cg1 coronal section immunostained for SK2 channel. SK2-positive cells are more prominent in L5 as opposed to L2/3. Scale bar, 20 μm. **b** Representative voltage traces of action potentials and medium afterhyperpolarization potentials (mAHPs) elicited by a 400 pA current injection under baseline conditions (black) and with 30% $N_2O$ (blue). Inset shows magnified mAHP as indicated by the teal box. **c** Summary data of mAHP amplitude across different current injection intensities (200–400 pA) under baseline (black) and 30% $N_2O$ (blue) conditions ($n = 9$ neurons from 4 mice). Two-way, repeated-measures ANOVA shows a main effect of $N_2O$ on the amplitude of mAHP ($F_{(1,8)} = 26.43$, $P < 0.001$). Sidak's post-hoc test results for individual current injections are shown in figure. **d** Representative control voltage traces under baseline conditions (black) and with 30% nitrogen ($N_2$) (magenta). **e** Summary data of mAHP amplitudes across different current injection intensities under baseline (black) and 30% $N_2$ (magenta) conditions ($n = 7$ neurons from 3 mice). Two-way ANOVA shows no effect of $N_2$ on mAHP amplitude ($F_{(1,6)} = 0.279$, $P = 0.616$), ns: $P = 0.616$. Data are presented as mean ± SEM.

bath (30%) induced significant reduction in mAHP amplitude across all tested current injection intensities as compared to baseline measurements (Fig. 6b, c). To ensure that the observed effects were specific to $N_2O$ and not the result of non-specific actions of gas application, we conducted control experiments using 30% nitrogen ($N_2$). As shown in Fig. 6d, e, $N_2$ application did not alter mAHP amplitudes across any of the tested current intensities. These results demonstrate that $N_2O$ robustly reduces mAHP in cortical L5 pyramidal neurons.

Next, using the approach detailed in Fig. 3, we explored whether local pharmacologic inhibition of SK2 channel function with specific inhibitor, apamin, could drive the spontaneous activation of neurons expressing SK2. Here, L5 and VIP neurons, but not L2/3 pyramidal cells or interneurons (PV or SST), were spontaneous recruited by apamin (Fig. 7a, b). NS 8593, a selective SK2 negative modulator, drove similar L5 responses (Fig. 7b). The spontaneous recruitment of L5 even occurred in the presence of local NMDA-R blockade with D-APV, confirming the NMDA-R-independent nature of this process (Fig. 7b). In contrast, local application of CyPPA, a SK2 channel activator, prior to $N_2O$ exposure reduced $N_2O$-induced L5 activity (Supplementary Fig. 15a). Like $N_2O$, a single apamin i.p. injection induced rapid antidepressant-like response in stressed mice (Fig. 7c).

Consistent with these pharmacologic manipulations, SK2 channel overexpression in L2/3 cell types, which normally do not express SK2 (i.e. pyramidal cells, PV, SST), enable acute $N_2O$ activation (Fig. 7d, e, Supplementary Fig. 15b). Knockdown of endogenous SK2 expression in L5 using Rbp4-Cre and Cre-dependent AAV encoding shRNA for KCNN2 prevented $N_2O$-induced L5 activation and its ensuing rapid

antidepressant-like response (Fig. 7d–f, Supplementary Fig. 15c, d). Collectively, these experiments suggest SK2 channel inhibition is necessary and sufficient to reproduce $N_2O$'s effect on both L5 and VIP cell function and drive its antidepressant-like response.

## $N_2O$-induced SK2 channel inhibition may arise via channel pore blockade

We hypothesized that $N_2O$ might inhibit channel function by interacting with its selectivity filter, anticipating that the doses of $N_2O$ required for clinical effect would allow enough $N_2O$ molecules to bind in the filter and yield a response, even if a large energetic barrier must be traversed. To evaluate this idea, we performed preliminary all-atom molecular dynamics (MD) simulation of an SK2 homology model with a single $N_2O$ molecule placed manually in its selectivity filter. In equilibrium MD simulation, the $N_2O$ molecule remained trapped in the selectivity filter with no migration (Supplementary Fig. 16a–c). To quantify the energetic barriers to $N_2O$ escape, we calculated a potential of mean force (PMF) profile of free energy of $N_2O$ diffusion along the pore axis in the region of the selectivity filter using the adaptive biasing force (ABF) method[53] which would allow recovery of a free energy profile across large energy barriers. The PMF profiles from replicate ABF simulations revealed that displacing the bound $N_2O$ 2 Å or more to approach either opening of the filter requires traversing an energy barrier of at least 8 kcal/mol (Supplementary Fig. 16d). Because the SK2 structure was a homology model, we chose not to exhaustively evaluate the numerous diffusion pathways available to $N_2O$ moving intracellularly away from the selectivity filter (regions with larger errors in Supplementary Fig. 16d) and therefore we draw no conclusions from

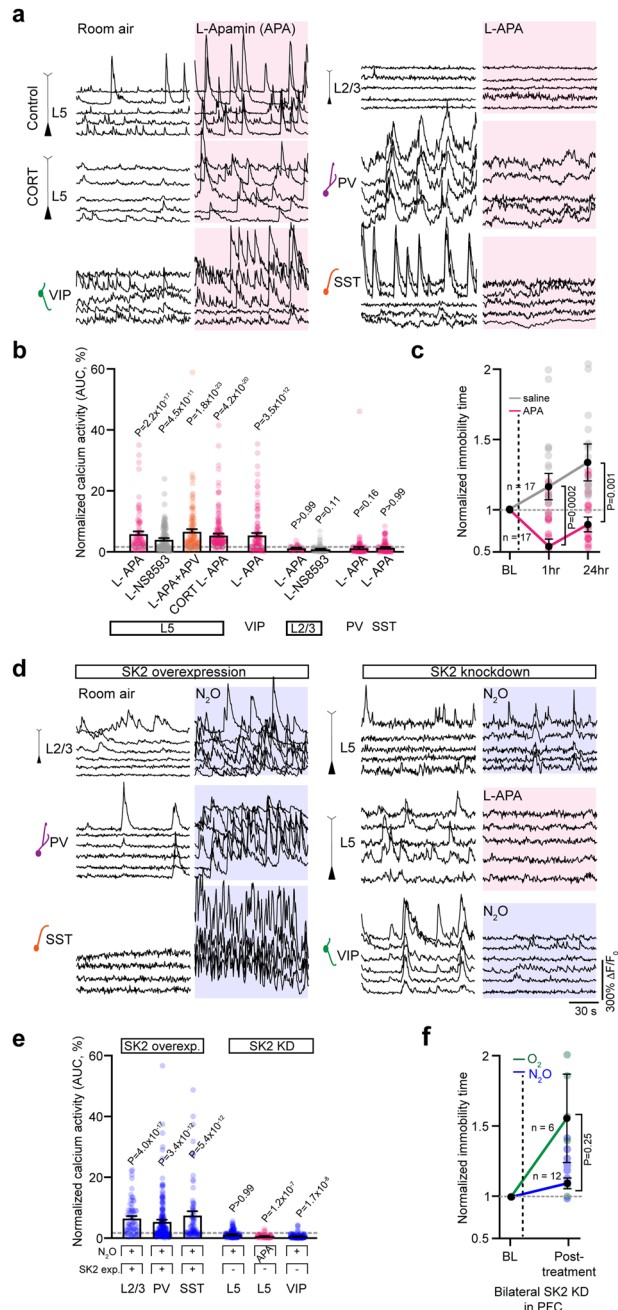

**Fig. 7 | Pharmacological inhibition of SK2 channels drives rapid L5 and VIP activation and antidepressant-like response. a** Representative GCaMP6 traces of individual L5 neurons under room air followed by local (L-) application of apamin to Cg1 (L-APA (magenta shaded region), 100 uM, 1 uL). Specific cell type noted left of traces. L5 and VIP neurons are spontaneously activated by L-APA whereas L2/3 pyramidal cells, PV, and SST cells fail to be recruited by L-APA. **b** Summary of drug-induced inhibition of SK2 responses across different neuronal cell types (Kruskal-Wallis (455): $P = 2.0 \times 10^{-92}$ followed by Dunn's multiple comparisons, L5 APA, $n = 90$ from 3 mice, $P = 2.2 \times 10^{-17}$; L5 NS8593, $n = 114$ from 3 mice, $P = 4.5 \times 10^{-11}$; L5 APA/APV, $n = 110$ from 3 mice, $P = 1.8 \times 10^{-23}$; CORT-treated L5 APA, $n = 145$ from 4 mice, $P = 4.2 \times 10^{-20}$; VIP APA, $n = 114$ from 3 mice, $P = 3.5 \times 10^{-12}$; L2/3 APA, $n = 105$ from 4 mice, $P > 0.99$; L2/3 NS8593, $n = 93$ from 3 mice, $P = 0.11$; PV APA, $n = 138$ from 3 mice, $P = 0.16$; SST APA, $n = 126$ from 5 mice, $P > 0.99$). **c** TST immobility time of CORT-treated mice injected with APA (0.1 mg/kg) or saline. APA-injected mice ($n = 17$) were significantly different from saline ($n = 17$) at 1 h post injection and 24 h later (two-sided Mann Whitney rank sum: 1 h, $P = 0.0002$; 24 h, $P = 0.001$). **d** Left, individual GCaMP6 traces of L2/3, PV, SST neurons overexpressing SK2 showing effect of $N_2O$ (blue shaded region). Right, effect of $N_2O$ or APA (magenta shaded region) on L5 and VIP neurons expressing SK2-shRNA. **e** Summary of SK2 over-expression and knockdown effects across different neuronal cell-types (Kruskal-Wallis (487): $P = 5.9 \times 10^{-102}$ followed by Dunn's multiple comparisons, SK2 over-expression in L2/3, $n = 107$ from 4 mice, $P = 4.0 \times 10^{-17}$; PV, $n = 228$ from 5 mice, $P = 3.4 \times 10^{-12}$; SST, $n = 86$ from 5 mice, $P = 5.4 \times 10^{-12}$ versus SK2 knockdown in L5, $n = 151$ from 4 mice, $P > 0.99$; L5 APA, $n = 170$ from 4 mice, $P = 1.2 \times 10^{-7}$; VIP, $n = 118$ from 4 mice, $P = 1.7 \times 10^{-8}$). **f** CORT-treated mice with bilateral expression of SK2-shRNA ($n = 12$) exposed to $N_2O$ fail to decrease their immobility time in response to $N_2O$ exposure as compared to $O_2$ ($n = 6$; two-sided Mann Whitney rank sum: $P = 0.25$). Data are presented as mean ± SEM.

these regions of the PMF. Overall, these data support the hypothesis that $N_2O$ may attenuate SK2 channel function in the activated state by becoming trapped in its selectivity filter[47]. Moreover, SK2 with stably trapped $N_2O$ likely contributes to the experimentally observed persistent L5 activity following $N_2O$ discontinuation.

## Discussion

While $N_2O$ has shown therapeutic promise for severe TRD, its molecular and circuit mechanisms of action are unknown. Using in vivo calcium imaging across Cg1 cortical layers and cell types in mice exposed to chronic stress, we show that subhypnotic $N_2O$ induces rapid and specific activation of L5 neurons that persists long after $N_2O$ clearance from the animal. L5 activation was crucial for rescuing stress-induced circuit hypoactivity in both L5 and L2/3 and driving a rapid antidepressant-like responses. Because synaptic loss and impaired connectivity are key features of chronic stress and depressive states in rodents and humans[4,8–10], these results provide a tangible mechanism

for how a single drug treatment can re-awaken existing dysfunctional circuits without the formation of new synaptic connections and contribute to rapid changes in behaviors (Supplementary Fig. 8d). The formation of synapses in response to $N_2O$ could contribute to stability of L5 and/or L2/3 activity and maintain these improvements over days. The extent and timescale over which $N_2O$-induced activity modulates new synapse formation requires future studies.

Our studies indicate that $N_2O$ has important mechanisms of action in cortical circuits beyond those observed in hippocampus and other brain regions. In particular, NMDA-R inhibition does not contribute to $N_2O$-induced cortical L5 activity patterns as evidenced by (1) $N_2O$-induces L5 activity in presence of NMDA-R blockers or GRIN1/NR1 knockdown (Fig. 3), (2) ketamine and $N_2O$ differentially modulate the same population of L5 cells when the drugs given in succession (Fig. 3), (3) subcellular imaging at the level of synapses and dendritic branches reveals no upregulation of synaptic inputs or dendritic calcium spikes (Fig. 4), and (4) dendritomies of apical dendritic branches from individual cells failed to abolish $N_2O$-induced L5 activity (Fig. 4). Although $N_2O$ can recruit L5 neurons even under conditions of low NMDA receptor activity, the sustained L5 activity and neuroplasticity following $N_2O$ elimination likely requires NMDA receptor signaling (Supplementary Fig. 8)[54]. This is in line with recent findings suggesting that ketamine may also depend on NMDA receptor signaling to produce its antidepressant-like behavioral effects[55].

In further search of an activating mechanism, we evaluated a series of molecular targets central to L5 excitability (Supplementary Figs. 11–12). Enhanced GABAergic tone via inhalation of subhypnotic concentrations of isoflurane or local application of muscimol eliminated $N_2O$-induced L5 activity (Supplementary Fig. 12). Calcium imaging of distinct GABAergic interneuron types revealed that $N_2O$ engages a specific disinhibition circuit via VIP-expressing cell recruitment with downregulation of PV- and SST-expressing neuronal activities (Fig. 5, Supplementary Fig. 8d). This shift in the inhibitory interneuron network favoring pyramidal cell excitation was necessary for $N_2O$-induced effect on L5 activity and behavior (Fig. 5). Thus, acute modulation of GABAergic interneurons by $N_2O$, like subhypnotic

ketamine, yielding a disinhibited cortical circuit might represent a unifying circuit phenotype to explain how these anesthetics enact their rapid-acting antidepressant action[39,56–58]. Dysfunctional stress-sensitive circuits could require transient drug-induced changes in inhibition to allow selective patterns of excitatory activity to propagate through cortex and engage rapid and durable forms of activity-dependent synaptic plasticity (Supplementary Fig. 8d)[59].

Despite emergence of a disinhibited network, we reasoned that $N_2O$-induced cell specificity must arise via an unknown molecular target, unique to L5 neurons and VIP interneurons. We identified SK2 channels as an attractive candidate with predominant L5 expression and sparse expression in L2/3. $N_2O$, but not $N_2$ gas, inhibited SK2-mediated mAHPs in vitro (Fig. 6). Local pharmacological inhibition of SK2 in Cg1 drove spontaneous activity in L5 and VIP, but not other cell types, and a $N_2O$-like antidepressant response (Fig. 7). SK2 over-expression and knockdown experiments in various cell-types support the effect of $N_2O$ acting as a SK2 channel inhibitor. Furthermore, MD simulations advanced our biophysical theory of $N_2O$ acting as a blocker in the highly conserved SK2 channel selectivity filter, where $N_2O$ traverses a substantial energy barrier to enter or exit its selectivity filter and inhibit channel function – potassium ion efflux and hyper-polarization (Supplementary Fig. 16). Altogether, these findings indicate that $N_2O$-induced rapid and sustained restructuring of prefrontal L5 neuronal activity is crucial for its antidepressant action and that this effect involves novel and specific molecular actions in distinct cortical cell types.

## Methods

### GCaMP6 and chemogenetic DREADD expression

The study was approved by the University of Pennsylvania Animal Care and Use Committee (approved protocol no. 807237). All animals were treated in strict accordance with National Institutes of Health (NIH) and institutional guidelines, the United States Public Health Service Policy on Humane Care and Use of Laboratory Animals, and the Guide for the Care and Use of Laboratory Animals. All mice were maintained at the University of Pennsylvania Perelman School of Medicine John Morgan animal facility with controlled temperature and humidity conditions and had free access to food and water/CORT water.

The genetically encoded calcium indicator GCaMP6f was used for calcium imaging of pyramidal neurons and interneurons in the cortex. GCaMP6f expression was performed using intracranial AAV injections of neonates. Pyramidal neurons were labeled using recombinant AAV9-*CaMKII*-Cre (Addgene, 105558) and AAV9-*CAG*-FLEX-GCaMP6f (Addgene, 100835) or AAV9-*hSyn1*-FLEX-GCaMP6f (Addgene, 100833); 10–100 nL of AAV per mouse. For sparse labeling of pyramidal neurons, we performed serial dilutions of Cre virus in artificial cerebral spinal fluid (aCSF)/fast-green dye (Sigma Aldrich, F725) solution followed by mixing with GCaMP6f and tdTomato viruses. The final injected AAV mix contained 80 % GCaMP virus, ~17.5 % tdTomato virus, and ~2.5 % Cre virus/fast-green dye. A glass micropipette (Drummond, 50001001 × 10) was pulled and beveled. A plunger was lightly oiled and inserted into the micropipette to pull the AAV mix. Subsequently, pups at postnatal day 1–2 were anesthetized by hypothermia (typically ~2–3 min on ice) and the micropipette was used (freehand) to penetrate the skin and skull and deliver ~200 nL of the virus mix. Medial prefrontal cortical (PFC) injections site was determined using landmarks—skull suture lines and head veins—as reference points (Supplementary Fig. 3). In Figs. 1 and 5, GCaMP6f expression was driven using AAV in *Rbp4*-Cre (GENSAT project at Rockefeller University, 031125), *Tlx3*-Cre (GENSAT; PL56), *Colgalt2*-Cre (GENSAT; NF107), *PV*-IRES-Cre (Jackson Laboratory, 008069), *Sst*-IRES-Cre (Jackson Laboratory, 013044) and *VIP*-IRES-Cre (Jackson Laboratory, 010908) mice using neonatal injection technique. Cre-expressing mice were genotyped for the presence of Cre based on established protocols as

an adult. Imaging was performed in 1 to 2-month-old mice, using both sexes, after at least 4 weeks of AAV expression. Mice were group-housed in temperature and humidity-controlled rooms on a 12-h light/dark cycle after injections.

**DREADD-induced modulation of cell types.** Layer 5 (L5) pyramidal neuron and cell type specific interneuron modulation was accomplished with expression of Cre-dependent DREADD-hM$_3$D(G$_q$) (AAV9-*hSyn1*-DIO-hM$_3$D(G$_q$)-mCherry; Addgene, 44361) or hM$_4$D(G$_i$) (AAV9-*hSyn1*-DIO-hM$_3$D(G$_q$)-mCherry; Addgene, 44362) under the human *synapsin-1* promoter in Cre-positive mice. For validation of L5 neuron or interneuron activation in vivo (Figs. 1j, 5d), two viruses (Cre-dependent GCaMP6f and DREADD-hM$_3$D(G$_q$)) were mixed at equal volumes and injected into the PFC of Cre-positive mice using neonatal injection technique, and following weeks of expression, L5 and inter-neuron activity was imaged before and ~20–30 min after CNO i.p. injection. In experiments where interneurons were activated and L5 pyramidal neurons were imaged (Fig. 5e), AAV1-*hSyn1*-DIO-hM$_3$D(G$_q$)-mCherry was mixed with AAV1-*hSyn1*-GCaMP6f, and AAVs were injected into interneuron-specific Cre-positive mice as a neonate. GCaMP6f expressing neurons in L5 confirmed based on cortical depth from pial surface.

**SK2 overexpression and knockdown.** SK2 channel overexpression in cells that normally lack endogenous SK2 expression was accomplished with expression of Cre-depedent mKCNN2 under the human *synapsin-1* promoter in Cre-expressing cells or Cre-positive mice (AAV9-*hSyn1*-DIO-mKCNN2-2A-mCherry; Vector Biolabs; reference sequence: NM_001312905; titer: $8.5 \times 10^{12}$ GC/mL)(Fig. 7c, d). SK2 channel knockdown in vivo and in vitro was accomplished by expression of Cre-dependent mKCNN2-shRNAmir under the *CAG* promoter in Cre-expressing cells or Cre-positive mice (AAV9-*CAG*-DIO-GFP-mKCNN2-shRNAmir#1; Vector Biolabs; reference sequence: NM_080465; titer: $6.4 \times 10^{12}$ GC/mL) (Fig. 7c–e, Supplementary Fig. 15). Based on shRNAmir#1 (5′- GCT GTCCATGTGAACGTATAATTCCGTTTTGG CCAC TGACTGACGGAATTATGTTCACATGGA CAG-3′) screening, shRNA-mir#1 yields 83% knockdown of mRNA (communication from Vector Biolabs). We confirmed this knockdown efficacy in cultured cortical neurons by measuring SK2 immunofluorescence in cells with or without shRNAmir#1 expression (~68% reduction of SK2 immuno-fluorescence in GFP-mKCNN2-shRNAmir#1 expressing cells as compared to GFP-scramble shRNA). Immunostaining protocol stated below with culture protocol.

**NR1 knockdown.** NMDA-receptor NR1 subunit knockdown in vivo and in vitro was accomplished by packaging GRIN1 siRNA oligos in AAV9 (Abmgood; Cat. No. 22671174; reference sequence: NM_008169) (Fig. 3c, e, Supplementary Fig. 8). A scrambled siRNA AAV was used in parallel for control (Abmgood, 01509). Based on target sequences (Target a - 7 ACCATGCACCTGCTGACATTCGCCCTGCT, Target b - 969 GGTGCTGATGTCTTCCAAGTATGCAGATG, Target c - 1507 GGA-GAGCTGCTCAGTGGTCAAGCAGACAT, Target d - 2591 ATAGAAA-GAGTGGTAGAGCAGAGCCCGAC) siRNA oligos expression should produce a 70% knockdown of mRNA (communication from Abm-good). We confirmed this knockdown efficacy in cultured cortical neurons by measuring NR1 immunofluorescence in cells with siRNA to GRIN1 as compared to scrambled siRNA control (~50% reduction of NR1 expression) (Supplementary Fig. 8).

### Chronic stress exposure

**Chronic Corticosterone (CORT).** Mice were exposed to CORT in the drinking water for 21 days[29,60]. CORT (Sigma-Aldrich, 27840) was dis-solved in limiting amounts of 100% ethanol and mixed with animal facility-provided drinking water to a final concentration of 0.1 mg/ml CORT and 1% ethanol. Both males and females were used for behavior

and imaging experiments but too few for statistical analysis of sex effect.

**Chronic aggressor interactions (CAI).** Male C57BL/6 mice were chronically stressed with a screened aggressor male CD-1 mouse (Charles River Laboratories, <4 months of age) for 10 min daily for a total of 10 days in CD-1's home cage (Supplementary Fig. 1)[61]. Females are not used in this model because prior attempts to perform defeat stress with C57BL6 among females did not generate aggressive interactions[62]. CD1 mice are housed singly throughout (exception was during pairing with C57BL/6 mice). To determine an aggressive CD-1 mouse prior to chronic stress, we placed screener C57BL/6 mice (different from stressed cohort) directly into the home cage of the aggressor for 180 s with the aggressor present. During three 180 s screening sessions, once daily, the CD-1 mouse had to attack in at least two consecutive sessions, with a latency to initial aggression <60 s. CD-1 mice that did not meet this criterion were not used. Our control group had normal drinking water and received gentle handling for ~15 min per day over 10 days prior to imaging.

### Surgical preparation before in vivo imaging
In preparation for imaging, mice underwent a surgical procedure to attach a head holder mount and create an imaging window for two-photon microscopy. In brief, mice were anesthetized with a mixture of 100% oxygen at 2 L min−1 and 1–4% isoflurane. A heating pad was used to maintain the animal's body temperature at approximately 37 °C. The mouse's head was shaved, and its skull surface was exposed with a midline scalp incision. The periosteal tissue over the skull surface was removed without damaging the temporal and occipital muscles. A head holder consisting of two parallel metal bars was attached to the animal's skull. In Cre-positive mice injected with AAV, <1% of mice were negative for GCaMP/DREADD-mCherry fluorescence, suggesting that genotyping error or off-target injection was a rare event. If mice were negative for expression, mice were euthanized and not used. In positive mice, a small skull region (~2–4 mm in diameter) located over interfrontal suture was removed (or parietal bone for S1), and a round glass coverslip (approximately the same size as the bone being removed) was affixed to the skull with Loctite 495 followed by dental acrylic cement. This window enabled imaging of PFC (+0.5–1.0 mm anterior of bregma and 0.3–0.5 mm lateral to midline), M2 (+1.5–2.00 mm, +0.5–0.8 mm) or S1 (−0.1 mm, +2.0 mm).

Upon recovering from surgical anesthesia, mice with head mounts were habituated daily (two sessions of 30 minutes with 15-min break) starting on postoperative day 1 in a custom-built body support to minimize potential stress effects of head restraining and imaging. No obvious distress was observed in habituated animals during imaging experiments. Mice tolerated surgery and stress related to the perioperative period as indicated by a 0–10% drop in weight. Imaging experiments were started on postoperative day 2–3 after window implantation.

### Two-photon calcium imaging.
On the day of imaging, awake mice were positioned in the custom head holder device under the two-photon microscope. In vivo two-photon imaging was performed with an Olympus DIY RS two-photon system (tuned to 910–920 nm) equipped with a Coherent Discovery NX laser. We minimized movement associated image artifact by head (secured metal head bars) and body (with a plastic sleeve) restraint on the imaging platform. Mice were head-restrained and imaged for <1.5–2 h in total, imaging across several regions in L2/3 or L5. Mice received one dose of N₂O (only exception was in Fig. 3 where ketamine was given) because of the potential interference of potentially N₂O-induced activity dependent plasticity. Calcium imaging was performed before and after 15–20 min of N₂O inhalation to increase the likelihood that the recordings were

taken under a steady state concentration. Multiple imaging planes were taken across the cortical mantle with a typical acquisition of 3–4 planes in a given cortical layer after this 20-min mark of inhalation. In Fig. 2, where the same mice were re-imaged hours later, the mice were placed back in their home cage with unlimited access to food and water. Pyramidal neurons and interneurons in cortical regions were randomly chosen and recorded for 2-minute sessions under awake conditions. Mice received N₂O (25, 50, 75 %; Airgas) mixed with O₂ (blended by Matrx MDS VMC anesthesia machine) or pure O₂ (100%) via nose cone under the two-photon microscope, and the same cortical regions were reimaged based on blood vessel maps. Exact N₂O concentration delivered post blending was monitored by a clinical Philips gas monitoring system (IntelliVue G5-M1019A/MP70; Medical grade device calibrated by Penn Clinical Engineering) (Fig. 1a). All experiments were performed using a ×20 Olympus objective (XLUMPLFLN; 1.00 NA, 2.0 mm working distance) immersed in aCSF, with ×2–6 digital zoom. Images were acquired at a frame rate of 2–4 Hz (2-μs pixel dwell time). Image acquisition was performed using Olympus Fluoview software and analyzed post hoc using ImageJ software version 2.1.0.

### Two-photon laser cutting of L5 tuft dendrites.
In Fig. 4 and Supplementary Fig. 9, dendrites expressing GCaMP6/tdTomato were cut by parking the two-photon laser beam on a small ROI that spanned the diameter of the dendrite for 5 s (tuned to ~890 nm and power was increased gradually until a sudden increase in fluorescence intensity was observed)[41]. After laser cut, a physical break could be observed between the two segments of dendrite. In the control experiment, the laser beam was parked ~30 μm away from the branch of interest without damaging it.

### Systemic and local drug delivery.
Drugs delivered systemically were via a single i.p. injection: CNO 3 mg/kg (Sigma-Aldrich, C0832; solution of CNO in saline 0.3 mg/mL), ketamine 10 mg/kg or 100 μM (Sigma-Aldrich, 1356009), fluoxetine 10 mg/kg (Cayman, 14418), and naloxone 5 mg/kg (Cayman, 15594). In Supplementary Fig. 12, mice were coadministered N₂O with isoflurane (Baxter, 0.2 or 0.6%). Imaging was performed after 15 min of administration to ensure a steady state concentration in the animal.

Drugs of various concentrations and low injectate volumes (~1 μL) were delivered locally, MK801 at 10–100 μM (Sigma-Aldrich, M107); D-APV at 100 μM (Tocris, 0106); CNQX at 100 μM (Sigma-Aldrich, 115066-14-3); Apamin at 100 μM (Tocris, 1652); NS8593 at 100 μM (Tocris, 4597); TTX at 1 nM (Tocris, 1069); riluzone 10 μM (Cayman, 35833); Cadmium at 50 μM (Sigma-Aldrich, 265330); Dantrolene at 500 μM (Cayman, 14326); Xestospongin C at 500 μM (Cayman, 64950), CyPPA at 100 μM (Cayman, 15614), muscimol at 10 μM (Sigma, 5060440001) via pressure application with a Picospritzer (20 p.s.i., 50 ms per pulse, 1 Hz, 5–10 pulses) to the surface of the PFC after removing a small bone flap (~200 μm in diameter) adjacent to the imaging window. The bone flap for drug delivery was made during head holder mounting and covered with a silicone elastomer such that it could be easily removed at the time of imaging. In some experiments, drugs were injected with Rhodamine 6 G to measure the extent of spread within cortical tissue (Fig. 3a). As a control, we applied aCSF after removing the bone flap.

### Cortical cell culture
Mouse cortical neurons were prepared by Penn Medicine Translational Neuroscience Center (Neurons R US) and plated at a low density in a similar manner to our previous report[63]. Culture medium contained 500 mL of MEM (Invitrogen), 5% FBS (HyClone, Logan, UT), 10 mL of B-27 supplement (Invitrogen), 100 mg of NaHCO3, 20 mM D-glucose, 0.5 mM L-glutamine, and 25 units/ml penicillin/streptomycin. Neurons were maintained at 37 °C in a 5%

$CO_2$-humidified incubator. AAV infection of GCaMP6f or shRNA/siRNA was performed on day 2 (2 days in vitro or DIV). Two-photon imaging of cultured neurons was performed on 7-9 DIV in a temperature-controlled chamber where neurons were recorded under normal bath solution followed by local application of glutamate 100 µM (Sigma-Aldrich, G1251) or bubbled $N_2O$ bath solution ($N_2O$ 50% mixed with $O_2$). For immunostaining of SK2 or NR1, cultured neurons infected with shRNA or siRNA respectively were rinsed with PBS twice and fixed for 15 min in a solution of 4% paraformaldehyde, pH 7.4. Coverslips were then rinsed three times in PBS. Cover slips were permeabilized for 1 h at room temperature with blocking buffer (10% NGS, 0.5% Triton X in 0.1 M PB). Primary antibodies were diluted in the blocking buffer and set to incubate overnight at 4 °C. SK2 was immunostained with rabbit monoclonal antibody (1:200; Boster Biological Technology, A05055). NR1 was identified with a NMDA-receptor 1 mouse monoclonal antibody (1:100; Fisher Scientific, 30-050-0). The next day, coverslips were rinsed three times in 0.1 M PBS for 10 min. Secondaries anti-mouse Cy3 or anti-rabbit Cy3 (Jackson ImmunoResearch) were diluted in blocking buffer and the coverslips were incubated for 2 hours at room temperature. Cover slips were rinsed four times in 0.1 M PBS for 10 min and mounted with Vectashield (Vector Laboratories) on glass slides.

### In vitro L5 recordings of mAHPs

Wild-type male mice aged 4–8 weeks were used for electro-physiological studies. Coronal brain slices (300 µm thick) were prepared as described previously[64]. Slices were perfused (2 ml/min) with oxygenated, artificial cerebrospinal fluid (aCSF) containing (in mM): 125 NaCl, 25 glucose, 25 $NaHCO_3$, 2.5 KCl, 1.25 $NaH_2PO_4$, 2 $CaCl_2$, and 1 $MgCl_2$; 310 mOsm; pH 7.3–7.4, equilibrated with 95% $O_2$/5% $CO_2$. Layer 5 pyramidal neurons were identified by location and morphology (large cell body with prominent apical dendrite). Whole-cell recordings were performed using borosilicate glass pipettes (World Precision Instruments, 3-6 MΩ) filled with K-MeSO$_4$ internal solution containing (in mM): 130.5 K-MeSO$_4$, 10 KCl, 7.5 NaCl, 2 $MgCl_2$, 10 HEPES, 0.1 EGTA, 2 MgATP, and 0.2 NaGTP; 290 mOsm; pH 7.3–7.4. mAHPs were recorded in current-clamp mode. Baseline recordings commenced 5 min after break-in to ensure stabilization. For $N_2O$ experiments, aCSF containing 30% $N_2O$ was prepared by mixing aCSF equilibrated with 95% $N_2O$/5% $CO_2$ and aCSF equilibrated with 95% $O_2$/5% $CO_2$. This solution was perfused for 5 minutes before recordings. Amplitude of mAHP was measured using Clampfit (Molecular Devices, San Jose, CA). Graphs and statistical analysis were performed in GraphPad Prism (Dotmatics, Boston, MA).

### Data analysis

**Behavior tests.** Behavior testing was conducted 30–60 min after post-$N_2O$ therapy. In tail suspension test (TST; Fig. 1b, Supplementary Fig. 1)[65], mice were suspended from the edge of a table (60 cm high) by an adhesive tape placed -1 cm from the tip of the tail. In TST, mice transition between complex escape movements, such as swinging and curling, mixed with periods of immobility over 6 min of video recording (first two minutes are discarded because of increased activity when placed in tail suspension). Videos and measurements were made using ANYMaze software (Stoelting).

In the elevated plus maze (EPM)[66], mice were placed in the center of the maze, which consisted of two open arms without walls and two closed arms with walls. The time spent over 10 min recording period in each arm of the maze was recorded using a camera suspended on scaffold and analyzed using ANYMaze. The maze was cleaned with an alcohol solution between trials.

In the sucrose preference test[67], mice were individually housed and given two bottles of tap water for 2 days. On day 3, once water intake was stable, one bottle was swapped for 0.6% sucrose water and volume intake was recorded for the next 12 h. The position of the sucrose bottle was randomly shuffled every 2 h. Preference for sucrose was calculated as the volume of sucrose consumed minus the volume of water, divided by the total intake volume, yielding a ratio from −1 to +1. Positive score indicates a sucrose preference, negative score a water preference, and a zero score suggests no preference.

To determine the effect of 50% $N_2O$ on animal behavior in the absence of head-restraining we created a closed chamber (12 × 7 x 8 inches; -11 liters) with a port to receive blended $N_2O$/$O_2$ (Supplementary Fig. 2). Spontaneous movement was first tracked in a pure $O_2$ environment for 5 min. Following this period, the chamber was transitioned to $N_2O$ 50% (2 L/min) for a period of 15 min to enable equilibrium prior to another 5 min recording. Total distance, average speed, and max speed over 5 min period was recorded by ANYMaze during both conditions.

**EEG recordings.** Epidural EEG leads were placed in M2 and RS cortex (0.3 mm and 1.6 mm posterior to bregma, 0.65 mm lateral to bregma). In Supplementary Fig. 5, EEG signals were recorded over wakefulness and increasing $N_2O$ (25, 50, 75%) at 1,000 Hz using a 32-channel headstage (Intan Technologies) using methodology previously described[68,69]. Each step was recorded for 30 minutes, where the final 10 min were used for subsequent analysis to ensure steady state. Differential biopotentials were processed and analyzed in MATLAB (2020b, Mathworks) with the Signal Processing and Statistics and Machine Learning toolboxes and custom code to compute power spectra under baseline wakefulness and $N_2O$ conditions using a multi-taper method (15 tapers, 5 s non-overlapping windows)[70]. Error estimation was computed using bootstrap resampling with replacement (1000 bootstraps, across windows) to produce 95% confidence intervals. Deviations from baseline were computed by subtracting the mean baseline power spectrum from each spectral window after injection. EEG was bandpass filtered (6th order Butterworth) from 0.5 Hz–100 Hz before spectral estimation and normalized by total power for each spectral window.

**Two-photon in vivo recordings.** During recordings, motion-related artifacts were typically <2 µm. Vertical movements were infrequent and minimized by two metal bars attached to the animal's skull (described above) and a custom-built body support. All time-lapse images from each individual field of view were motion-corrected and referenced to a single template frame using cross-correlation image alignment (TurboReg plugin for ImageJ version 2.1.0). ROIs corresponding to visually identifiable somas (pyramidal cells and interneurons) were selected manually from the field of view. Imaging planes were acquired from L2/3 and L5, corresponding to cells positioned -150–350 µm and -500–750 µm from the pial surface respectively. Note that our neonatal injections produced sparse to moderate labeling of GCaMP6 throughout all layers in PFC. As one can see in Fig. 1, 5–20 neurons per L2/3 or L5 imaging region was common. A typical experiment would include 3–4 (randomly chosen) imaging regions per animal, yielding -20–40 cells per animal. Somas that could be identified in all imaging sessions were included in the dataset.

**mRNA expression analysis from Allen brain cell type specific dataset.** We queried the 10x scRNAseq whole brain dataset from Allan Brain Cell Atlas[45] to identify gene expression levels of candidate channels and receptors (Kcnq2, Kcna1, Hcn1, Scn1a, Scn1b, Scn2a, Scn3a, Cacna1c, Ryr1, Itpr3, Oprm1, Grin1, Kcnn1, Kcnn2, Kcnn3, Grm1) across select neuronal cell types (L2/3, L5 IT, L5 ET, VIP, SST, PV). We limited our search to include feature matrix labels from cortical regions only and cell types to be restricted to cortex as well. A total of 109,503 individual cells were included in this analysis (L2/3 = 49,196 cells, L5 IT = 19,905 cells, L5 ET = 5,990 cells, VIP = 11,857 cells, SST = 13,845 cells, VIP = 9,070 cells). Distributions of gene expression are shown as log2(CPM + 1) as violin plots, where black line denotes

means, red line denotes medians. CPM = counts per million (transcript reads). Using this screen, we identified the SK2 channel, encoded by the Kcnn2 gene, to be specifically enriched in Layer 5 ET (extra-telencephalic) and VIP neurons as compared to other cell types (L2/3, L5 IT, SST, PV) (Supplementary Fig. 14).

**Analysis of calcium signals.** In this study, we used GCaMP6f, an indirect reporter of neuronal spiking activity. All the pixels inside the ROI were averaged to obtain a fluorescence trace for each ROI. Back-ground fluorescence was calculated as the average pixel value per frame from a region without GCaMP expression (blood vessel) and subtracted from the time-series fluorescence traces. The baseline ($F_0$) of the fluorescence trace was estimated by the average of inactive portions of the traces (-2 seconds). We did not smooth the raw fluorescence trace (raw traces are presented throughout the manuscript in each figure). The $\Delta F/F_0$ (%) was calculated as $\Delta F/F_0 = (F - F_0) / F_0 \times 100$. Representative traces from a given cortical region are selected at random.

In some conditions, GCaMP6f can produce large fluorescence transients (~20% $\Delta F/F$) even in response to single action potentials, and individual spikes within a burst result in stepwise fluorescence increases[71]. However, when neuronal firing rates are high, it becomes difficult to resolve the number of action potentials owing nonlinear responses and to the long decay kinetics of GCaMP6 fluorescence. We found that there was a diversity in calcium traces of pyramidal cells in room air conditions and under $N_2O$, which likely reflects burst and nonburst firing of L5 neurons. To avoid assumptions related to spike inference and challenges that arise with GCaMP reporting of firing rates in bursting neurons, we performed an integrated measurement of a cell's output activity over 2 min recording, termed area under the curve (AUC, %), as well as measuring peak fluorescence signal.

The effect of either $N_2O$ or ketamine was defined as the difference in L5 $\Delta F/F_0$ in drug-induced state and wakefulness. In Fig. 3f where a scatter plot of calcium activity of individual L5 neurons are shown, we compute the Pearson correlation coefficient. The $P$ value associated with the correlation coefficient was computed by transforming the correlation coefficient to an $F$-statistic having $n-1$ and $n-2$ degrees of freedom, where $n$ is the number of neurons. Negative correlation coefficients in this case mean that $N_2O$ and ketamine activated neurons differently.

## SK2 channel molecular dynamics simulations

**System setup.** A homology model for human SK2 was constructed using SWISSMODEL[72] with open-state, calmodulin-bound human SK4 channel (PDB 6CNN) as a template structure[73]. The template structure has calcium and calmodulin bound, yielding an open channel state. SK2 sequences are 98.8% identical between human and mouse. The resulting highest-scoring homology model, with calmodulin in the same bound position and conformation as in the SK4 structure, was embedded in a POPC:cholesterol 3:1 lipid membrane, using CHARMM-GUI[74,75]. The channel was oriented with the major pore axis parallel to the simulation box z-axis.

**Simulations.** A single $N_2O$ molecule, using previously described molecular mechanics parameters[76], was placed in the SK2 channel selectivity filter. A volume restraint, upon which a restoring force would be imposed upon migration >8 Å away from the initial position, was placed on the $N_2O$ molecule. NAMD 3 was used for all simulations[77]. Minimization and equilibration followed. Production simulation in the isothermic-isobaric ensemble with Langevin dynamics (2 fs timestep, Particle Mesh Ewald used for long-range electrostatics with 10 Å switching distance and 12 Å cutoff) for 20 ns followed. As the ligand remained in the pocket without migration further than 8 Å, the volume restraint did no work. To generate a potential of mean force (PMF) profile of the movement of

$N_2O$ along the pore axis in the immediate region of the selectivity filter, we conducted adaptive biasing force simulations[78]. The position of the $N_2O$ molecule along the pore axis was measured using the Colvars module[76,79], as the z-component of the distance between the center of gravity of the $N_2O$ and the averaged center of mass of the C-alpha atoms of the Tyr361 residues (in the selectivity filter) in each of the 4 monomers comprising SK2. Data was collected in 0.1 Å bins. In each bin, 500 steps were discarded prior to collecting ABF data. Each ABF calculation spanned 6 Å along the pore axis. A total of 15 replicate calculations over a total of 135 ns were run.

## General statistical analysis

Animals were randomly assigned to experimental groups. No statistical methods were used to pre-determine sample sizes, but our sample sizes for in vivo imaging and behavior studies are similar to those reported in our previous publications and others[39–41,80,81]. The interventions were not blinded as nearly all experiments were carried out and analyzed by J.C. We tested the data for normality using the Shapiro-Wilk test and performed parametric statistical tests. Two-way ANOVA was used in Figs. 1b–d, k, 3c, e. If normality was not present, we performed nonparametric tests, including Wilcoxon rank sum test (or t test) to compare two groups and Kruskal-Wallis (or one-way ANOVA) to compare more than two groups. Kruskal-Wallis (or One-way ANOVA) tests were followed by Dunn's multiple comparisons test for multiple comparisons. Mean ± s.e.m. was used to report statistics unless otherwise indicated. The statistical test used and the definition of $n$ for each analysis are listed in the text or figure legends. Tests were computed in GraphPad Prism version 9.3.1. EEG analysis and mRNA expression plots were performed in MATLAB (2023a) with custom scripts. Criteria for animal exclusion was pre-established: mice were excluded if the injected virus did not express. Exact $P$ values are reported in figures and legends.

## Reporting summary

Further information on research design is available in the Nature Portfolio Reporting Summary linked to this article.

## Data availability

We declare that all data supporting the findings of this study are provided within the paper and its supplementary information. Underlying data of all figures are provided in the Source Data file with this paper and data are fully available from the corresponding author on request. Source data are provided with this paper.

## Code availability

No new code was generated in this work.

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

## Acknowledgements

We thank all members of the Cichon lab including Marie Fina for critical support in animal breeding and genotyping. We also appreciate Dr. Alex Proekt for his assistance in analyzing the expression of different RNAs of interest amongst various cortical cell types. Work supported by National Institutes of Health R35GM151160-01 and BBRF Young Investigator award to J.C., NIMH, American Foundation for Prevention of Suicide, Brain & Behavior Research Foundation to P.N., K08GM139031 to T.T.J., R01GM088156, R01 GM151556 to M.B.K and MH122379 to C.F.Z and S.J.M.

## Author contributions

J.C., M.B.K., and P.N. initiated the project. J.C. performed the experiments and analyzed the behavior and imaging data. X.L., S.M., C.Z. performed in vitro L5 recordings of mAHPs. T.T.J. performed molecular dynamics simulations on SK2. A.W. performed and analyzed the EEG and RNA expression data set from Allen Brain. J.C. wrote the paper with input from all authors.

## Competing interests

P.N. is currently receiving or has received funding from NIMH, American Foundation for Prevention of Suicide, and has received reimbursement as advisory board member for Becton-Dickinson unrelated to this work and has previously filed for intellectual property protection related to the use of nitrous oxide in major depression and is the co-founder of NitroTherapeutics, Inc., a company that aims to develop nitrous oxide as treatment for major depression. He has received remuneration from the American Society of Anesthesiologists for serving as editor in the journal Anesthesiology. C.F.Z. serves on the Scientific Advisory Board of Sage Therapeutics and has equity in the company. Sage Therapeutics was not involved in this work. The remaining Authors declare no competing interests.
