## [Transparent Peer Review file · Nature Communications]

Nitrous Oxide activates layer 5 prefrontal neurons via SK2 channel inhibition for antidepressant effect

Corresponding Author: Dr Joseph Cichon

A version of this paper was originally rejected for publication by Nature Communications, however that decision was reconsidered after appeal by the authors.

Version 1:

Reviewer comments:

Reviewer #1

(Remarks to the Author)

The authors did an excellent job addressing the comments including a number of new analyses, and some crucial new experiments such as slice electrophysiology shown in Figure 6. The edits to the main text helped clarifying many aspects of the paper. As noted before, there is interesting in the clinical use of nitrous oxide. This study that dissects the cortical circuit mechanisms underlying nitrous oxide administration will be of interest to the field.

One last comment – it would be very helpful to see a plot where y-axis is average dF/F or some normalized changes in neural activity, and x-axis is time on the scale of minutes, showing clearly when L5, L2/3 and the various interneurons types get recruited to be active after the onset of N_2O , and then which and for how long some of these activity persist. Right now most figures are showing simply “baseline” and “post N_2O ”, which does not give a sense of the timing of these neural activations.

Reviewer #2

(Remarks to the Author)

The authors have addressed my concerns except one: I would recommend putting the MD simulations in the supplement and label them 'preliminary'

Reviewer #3

(Remarks to the Author)

The authors have addressed the majority of my concerns with clarifications, edits, new analyses and substantial new data. The removal of prior data and addition of new behavioral data with good group sizes and simpler experimental designs has greatly clarified the anti-depressant effect of N_2O and supports the authors initial claims. The explanations concerning use of independent cohorts for different timepoints, increasing minimum group sizes and new statistical analyses have adequately addressed my concerns relating to imaging experiments. This is a rigorous and exciting paper that leverages an impressive combination of cutting edge techniques and thoughtful experimental approaches to introduce new insight and new ideas that are sure to be impactful in the field. Yet, one easily addressable point raised in my initial review remains to be satisfactorily addressed.

The authors should edit the manuscript to clearly indicate the sex of mice included in every experiment and, wherever it is feasible to show individual data points, color these to indicate the sex of the mouse/cell. This could either be overlaid in the plots in the main figure or added as separate plots to the supplementary figures. This is an easy change that is important to support the broader impact of this work. The authors have indicated that sample sizes are insufficient to conduct an analysis by sex and that is ok! Yet, as a minimum standard, reporting and visualizing the individual data points for males and females allows readers to interpret the extent to which these findings are generalizable to both sexes and the existence of potential sex differences to pursue in future research. In the context of depression and anti-depressant treatment, this is

particularly pertinent given known sex/gender differences in incidence, symptom profiles and treatment response in addition to increasing findings of robust sex differences in preclinical depression research. Following this simple revision, this work would be acceptable for publication.

Response in blue

Reviewer #1 (Remarks to the Author):

The authors did an excellent job addressing the comments including a number of new analyses, and some crucial new experiments such as slice electrophysiology shown in Figure 6. The edits to the main text helped clarifying many aspects of the paper. As noted before, there is interesting in the clinical use of nitrous oxide. This study that dissects the cortical circuit mechanisms underlying nitrous oxide administration will be of interest to the field.

One last comment – it would be very helpful to see a plot where y-axis is average dF/F or some normalized changes in neural activity, and x-axis is time on the scale of minutes, showing clearly when L5, L2/3 and the various interneurons types get recruited to be active after the onset of N₂O, and then which and for how long some of these activity persist. Right now most figures are showing simply “baseline” and “post N₂O”, which does not give a sense of the timing of these neural activations.

We appreciate the reviewer’s comment regarding the rapidity of nitrous oxide-induced effects on neuronal activity. In this manuscript, our focus was on capturing the steady-state effect (>15 min) when describing cortical circuit reconfiguration, rather than the dynamic changes occurring during the initial minutes of inhalation when end-site concentrations are still variable.

As requested by Reviewer 1, we have added new experiments (n = 3 animals) in **Supplemental Fig. 4g**, where we track cortical neuronal activity in layer 2/3 and layer 5 neurons at the onset of nitrous oxide inhalation (50%), rather than at steady-state concentrations as reported throughout the manuscript. These experiments reveal that nitrous oxide rapidly (within minutes) recruits L5 neurons but does not significantly affect L2/3 neurons. Notably, L5 activity appears to still be increasing at the conclusion of this recording.

Regarding the duration of this activity, we are planning follow-up studies to specifically address this question. More precisely, we aim to investigate:

1. The persistence of neuronal activity after nitrous oxide exposure.
2. The role of NMDA receptor function in sustaining this activity, as suggested by insights from **Supplemental Fig. 8a-b**.
3. The potential role of dendritic integration in maintaining this activity state.

We hope the reviewer understands our decision to leave this dataset for a separate, dedicated study that will further explore these mechanistic aspects in greater depth.

Reviewer #2 (Remarks to the Author):

The authors have addressed my concerns except one: I would recommend putting the MD simulations in the supplement and label them 'preliminary'

We have now moved the MD simulations to **Supplemental Fig. 16** and updated the text accordingly:

"To evaluate this hypothesis, we conducted preliminary all-atom molecular dynamics (MD) simulations of an SK2 homology model, manually placing a single N₂O molecule in its selectivity filter."

Reviewer #3 (Remarks to the Author):

The authors have addressed the majority of my concerns with clarifications, edits, new analyses and substantial new data. The removal of prior data and addition of new behavioral data with good group sizes and simpler experimental designs has greatly clarified the anti-depressant effect of N₂O and supports the authors initial claims. The explanations concerning use of independent cohorts for different timepoints, increasing minimum group sizes and new statistical analyses have adequately addressed my concerns relating to imaging experiments. This is a rigorous and exciting paper that leverages an impressive combination of cutting edge techniques and thoughtful experimental approaches to introduce new insight and new ideas that are sure to be impactful in the field. Yet, one easily addressable point raised in my initial review remains to be satisfactorily addressed.

The authors should edit the manuscript to clearly indicate the sex of mice included in every experiment and, wherever it is feasible to show individual data points, color these to indicate the sex of the mouse/cell. This could either be overlaid in the plots in the main figure or added as separate plots to the supplementary figures. This is an easy change that is important to support the broader impact of this work. The authors have indicated that sample sizes are insufficient to conduct an analysis by sex and that is ok! Yet, as a minimum standard, reporting and visualizing the individual data points for males and females allows readers to interpret the extent to which these findings are generalizable to both sexes and the existence of potential sex differences to pursue in future research. In the context of depression and anti-depressant treatment, this is particularly pertinent given known sex/gender differences in incidence, symptom profiles and treatment response in addition to increasing findings of robust sex differences in preclinical depression research. Following this simple revision, this work would be acceptable for publication.

We appreciate the reviewer's suggestion to clearly indicate the sex of mice in all experiments and agree that transparency in reporting is important for ensuring the broader impact of this work. While replotting all figures to overlay individual data points by sex was considered, we believe this would overly complicate the plots, making them less interpretable.

To address this concern while maintaining clarity, we have now denoted the sex of each data point in the source data Excel spreadsheet, which captures all data presented in the figures.

This allows readers to examine sex-based distributions and assess potential differences while keeping the main figures clear and focused.

We hope this approach meets the reviewer's request while preserving the readability of our figures. Thank you for the valuable suggestions, and we appreciate the constructive feedback in strengthening the manuscript.